# Floquet Conformal Field Theories with generally deformed Hamiltonians

Ruihua Fan[1], Yingfei Gu[3,1], Ashvin Vishwanath[1], and Xueda Wen[1,2]

[1]*Department of Physics, Harvard University, Cambridge MA 02138, USA*
[2]*Department of Physics, Massachusetts Institute of Technology, Cambridge, MA 02139, USA*
[3]*California Institute of Technology, Pasadena, CA 91125, USA*

February 9, 2021

## Abstract

In this work, we study non-equilibrium dynamics in Floquet conformal field theories (CFTs) in 1+1D, in which the driving Hamiltonian involves the energy-momentum density spatially modulated by an arbitrary smooth function. This generalizes earlier work which was restricted to the sine-square deformed type of Floquet Hamiltonians, operating within a $\mathfrak{sl}_2$ sub-algebra. Here we show remarkably that the problem remains soluble in this generalized case which involves the full Virasoro algebra, based on a geometrical approach. It is found that the phase diagram is determined by the stroboscopic trajectories of operator evolution. The presence/absence of spatial fixed points in the operator evolution indicates that the driven CFT is in a heating/non-heating phase, in which the entanglement entropy grows/oscillates in time. Additionally, the heating regime is further subdivided into a multitude of phases, with different entanglement patterns and spatial distribution of energy-momentum density, which are characterized by the number of spatial fixed points. Phase transitions between these different heating phases can be achieved simply by changing the duration of application of the driving Hamiltonian. We demonstrate the general features with concrete CFT examples and compare the results to lattice calculations and find remarkable agreement.

# 1   Introduction

Non-equilibrium dynamics in time-dependent driven quantum many-body systems has received extensive recent attention. Floquet driving sets up a new stage in the search for novel systems that may not have an equilibrium analog, e.g., Floquet topological phases [1–13] and time crystals [14–22]. It is also one of the basic protocols to study non-equilibrium phenomena, such as localization-thermalization transitions, prethermalization, dynamical Casimir effect, etc [23–31]. In this work, we are interested in a quantum $(1 + 1)$ dimensional conformal field theory (CFT), which may be viewed as the low energy effective field theory of a many-body system at criticality. The property of conformal invariance at the critical point can be exploited to constrain the operator content of the critical theory [32,33]. In particular, for $(1+1)$D CFTs, the conformal symmetry is enlarged to the full Virasoro symmetry, which makes it tractable in the study of non-equilibrium dynamics, such as the quantum quench problems [34,35].

There are several possibilities of choosing the driving protocols, such as locally coupling the system to an external source [36,37]. In this work, the protocol will be implemented with the deformed Hamiltonians

$$H_v = \int_0^L \frac{dx}{2\pi} \left( v(x)T(x) + \overline{v}(x)\overline{T}(x) \right), \tag{1}$$

where $v(x) = v(x+L)$ and $\overline{v}(x) = \overline{v}(x+L)$ are two independent smooth real functions, dubbed deformation functions. Here $T(x)$ and $\overline{T}(x)$ are the chiral and anti-chiral energy-momentum density, namely, $T + \overline{T}$ is the energy density and $T - \overline{T}$ the momentum density. The ordinary homogeneous CFT Hamiltonian corresponds to $v(x) = \overline{v}(x) = 1$. There are a few advantages to this choice. First, it does not rely on the operator content or the symmetry of the system and should lead to universal results. Second, the corresponding operator evolution has a particularly simple geometric interpretation, that deserves more explanation below. Moreover, it also has a natural generalization to higher dimensions, which will be explained in later sections.

The application of this setup to Floquet problems was initiated by Ref. [38], where the deformation is a sine-square function $v(x) = \overline{v}(x) = \sin^2(\pi x/L)$. This special choice has an underlying $SL_2$ algebraic structure that facilitates a thorough discussion on the dynamics, including the quasi-periodic and random driving cases [38–45]. In all three cases, both the non-heating and heating phases are identified, and the heating phase is found to exhibit energy-momentum peaks with linearly growing entanglement. In addition, a quasi-particle picture was also introduced to interpret the results above [40,43]. Namely, the energy and entanglement are assumed to be carried by fictitious quasi-particles, which see the sine-square function as their velocity profile and move accordingly. When their stroboscopic motion (i.e. observing their positions at the end of each cycle of driving) has stable fixed points, the quasi-particles will accumulate at those locations, which gives rise to the energy peaks and growing entanglement. This is the geometric interpretation of (1) advertised above phrased in terms of quasi-particle motion.

This work generalizes the discussion to arbitrary smooth $v(x)$ and $\overline{v}(x)$ in (1). Solving the dynamics from the algebraic viewpoint is no longer tractable because the underlying algebra becomes the infinite dimensional Virasoro algebra. On the other hand, the geometric viewpoint still works in this general case. Namely, the deformation functions $v(x), \overline{v}(x)$ can still be considered as the velocity of quasi-particles. In more technical terms, the operator evolution $e^{iH_v t}O(x)e^{-iH_v t}$ follows a conformal transformation, which is intimately related to the classical trajectory of quasi-particles generated by $v(x), \overline{v}(x)$. In this work, we will follow the geometric viewpoint exclusively and give quasi-particle interpretation to most of the results.

**Concrete examples** Having a less constrained deformation function $v(x)$ brings much more possibilities to the Floquet dynamics. To have an idea of what could happen, we construct a concrete example, the main result of which is shown in Fig. 1 and Fig. 2, which is explained briefly below: (More details are in Sec. 3.4)

1. There are still non-heating and heating phases, as shown by the yellow and blue regions in Fig. 1 (a) respectively. However, the heating phase regions are rather fragmented compared with the $SL_2$ case.

2. On the contrary to the $SL_2$ case, where all the heating phases have the same number of energy-momentum peaks, the heating phases in the general setup can have different number of peaks, denoted by the white and black text in Fig. 1 (a). Typical configurations of the energy-momentum peaks are shown in Fig. 1 (b). Notice that the energy momentum peaks do not necessarily have the same height, which is also different from the $SL_2$ case. After each cycle, those peaks can shuffle their position cyclically and come back to its original position after every $p$ cycles. In the case of Fig. 1 (b), let $x_{i=1,2,3}$ denote the positions of the peaks from the left to right. Then the peak at $x_1$ can move to $x_2$ after one cycle of driving and so on, then return to $x_1$ after 3 cycles. Fig. 1 (b) shows the results for every 6 cycles, which is why we do not see the position-switching from the figure. This micro-motion also manifests in the entanglement entropy growth as mentioned below in Point 4.

3. These energy-density peaks also share entanglement with and only with its nearest neighbors, which can be explained by the quasi-particle picture as follows (see Fig. 1(c) for a

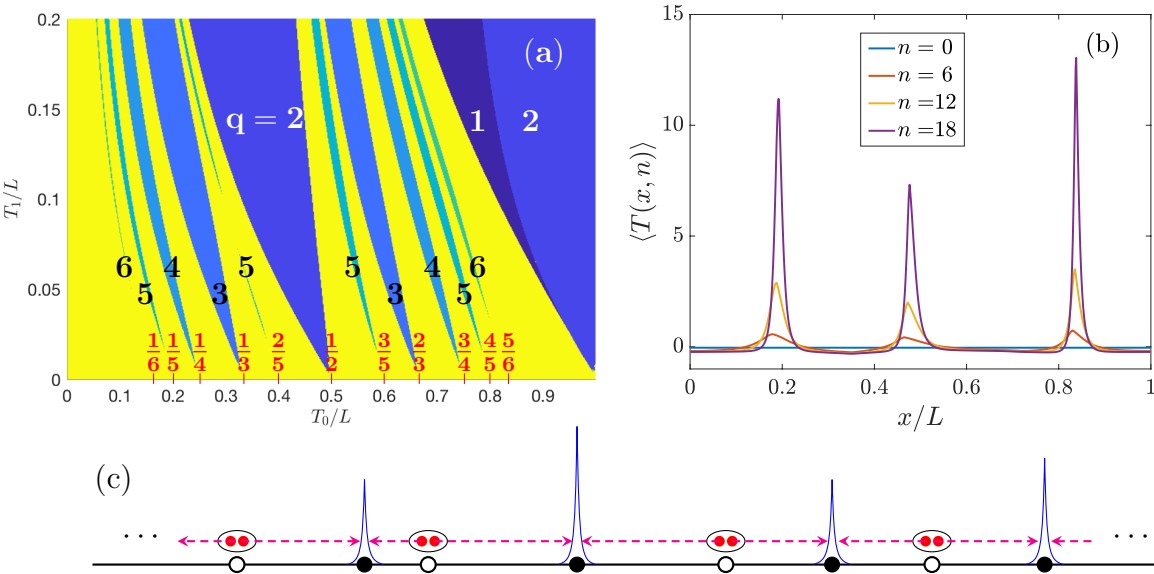

Figure 1: (a) Phase diagram in a generalized Floquet CFT, with both heating phases (in blue) and non-heating phases (in yellow). There are distinct heating phases characterized by different numbers ($q$) of fixed points in the operator evolution. (b) Time evolution of the chiral energy-momentum density in the heating phase with period-3 fixed points. (c) A cartoon plot of the emergent entanglement pattern and the energy-momentum density distribution (in real space) in the heating phase of a generalized Floquet CFT. During the driving, the unstable fixed points (∘) serve as sources of quantum entanglement. The degrees of freedom carrying quantum entanglement (pairs of •), which can be intuitively viewed as EPR pairs, flow from the unstable fixed points (∘) to the nearby two fixed points (•) separately. This process creates the entanglement between every neighboring peaks (blue spikes) of energy-momentum density.

cartoon). The initial state contains quasi-particles that are locally entangled with a partner, as denoted by the pairs of red dots, which move chirally under the time evolution. Namely, two particles in a pair move towards different directions. In the heating phase, their motions have both unstable (vacant dots) and stable fixed points (black dots), the former are the main source of particles and the later are the sink. These energy peaks can be understood as an accumulation of particles that come from the source, denoted by the red dots on top of the vacant dots. Since each dot is entangled with its partner who departures towards different destinations, this gives rise to the energy peaks as well as their entanglement pattern.

4. We also study the entanglement entropy for a fixed subsystem that encloses one or multiple peaks (if they exist). In the non-heating phase, the entanglement oscillates in time (yellow curve in Fig. 2). In the heating phase, the entanglement overall grows linearly (blue and red curve in Fig. 2), while it can exhibits oscillating behavior within small time scales. For example, the red curve shows a clear oscillation within every three cycles. This is a manifestation of the periodicity of the micro-motion of the energy-momentum peaks shown in Fig. 1(b).

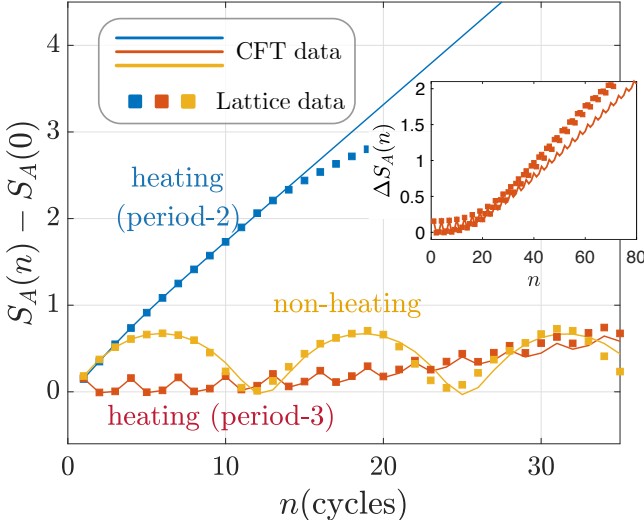

Figure 2: Time evolution of the entanglement entropy in the non-heating phase, and the heating phases with period-2 and period-3 fixed points, respectively. A longer time scale for the heating phase with period-3 fixed points is shown in the inset.

5. In addition to the CFT calculation, we also did a lattice simulation using complex free fermions with nearest neighbor hopping and find good agreement, as shown by the dots in Fig. 2.

**General features**   With this concrete example in mind, now we summarize the general features for Floquet CFT with generally deformed Hamiltonians:

1. The presence of fixed points determines the heating phase. There are generally $q > 1$ fixed points. However, the conformal map (quasi-particle motion) associated to the single-cycle Floquet driving may miss some spatial fixed points that only appear in multi-cycle driving. This phenomenon also exists in the simple $SL_2$ deformed Floquet CFTs [43]. After each driving cycle, these spatial fixed points will shuffle among each other in the array, and come back to the original locations after $p$ ($p \geq 1$) driving cycles, thus dubbed as period-$p$ fixed points. We will show that all the fixed points must share the same periodicity, which is then a good characteristic of the heating phase. Notice that the number of fixed points does not have to be equal to the periodicity.

2. These spatial fixed points, stable and unstable ones, determine the entanglement growth. When the subsystem encloses an unstable fixed point in the operator evolution, [1] the entanglement will grow in time. However, the growth rate is controlled by the nearby stable fixed points. Thus, it is tempting to interpret the unstable fixed point as the sink of EPR pairs that are created at stable fixed points. Therefore, the stable/unstable spatial fixed points determine the patterns of entanglement in the Floquet driving.

---

[1] As will be discussed in detail in Sec. 2, in the study of operator evolution as opposed to state evolution, the time direction is reversed. As a result, the stable/unstable fixed points for the quasi-particle moving (state evolution) in Fig. 1(c) will correspond to the unstable/stable fixed points in the operator evolution.

3. These spatial fixed points also determine the energy-momentum density distribution. We show that if the initial state is a highly excited state or a thermal ensemble in high temperature, then there must be peaks of energy-momentum density at the unstable spatial fixed points of operator evolution. If the initial state is chosen as the ground state of homogeneous CFT Hamiltonian, quantum fluctuation may cause subtleties and we refer readers to Sec. 3.3 for detailed discussion.

The structure of the rest of this paper is arranged as follows. In Sec. 2, we define "deformed Hamiltonian" in CFTs of general dimensions and specialize ourselves to (1+1)d to show the operator evolution as a conformal map. In particular, we explain how to interpret the deformation as a velocity to generate that conformal map. In Sec. 3, we discuss all possible Floquet dynamics with arbitrary smooth deformation with a minimal setup. The more familiar $SL_2$ case is first reviewed in Sec. 3.2 to prepare us with all the necessary concepts before the general discussion in Sec. 3.3 so that readers can explicitly see how they fall into a unified framework and how general deformation brings more possibilities to the phenomena. We also provide a specific example in Sec. 3.4 to demonstrate some of the features in our general discussion, which is further supported by a lattice simulation. Readers more interested in concrete models can directly jump to Sec. 3.4 after Sec. 3.2, and come back to the general discussion later. Finally, we conclude with a few discussion. Appendix. A concerns with a global quench problem, which is inspired by our discussion in Sec. 3.

# 2 Operator evolution under deformed Hamiltoninan

In this section, we start with the topological surface operators associated with the conformal symmetries in general dimensions, and show that the deformed Hamiltonian (1) belongs to such family when the spacetime dimension is two. Then we demonstrate that the operator evolution driven by (1) is expressible as a conformal transformation generated by the deformations $(v(x), \overline{v}(x))$ in the Hamiltonian. Finally, we apply the formalism to the entanglement and energy density calculation.

## 2.1 Conformal symmetry and the deformed Hamiltonian

A conformal field theory in the space-time dimension $d > 2$ is equipped with $SO(d+1,1)$ as its symmetry group[2]. Geometrically, the conformal group $SO(d+1,1)$ is specified by $\frac{(d+2)(d+1)}{2}$ conformal Killing vectors $\xi^\mu$, which generate the conformal transformations. In the operator language, these transformations can be realized by the following topological surface operators

$$Q_\xi(\Sigma) = - \int_\Sigma d\sigma^\mu \xi^\nu T_{\mu\nu} \tag{2}$$

where $\Sigma$ is a codimension-1 surface, and the operators are conserved/topological in the sense that they are invariant under small deformation of $\Sigma$, i.e. the surface $\Sigma$ can shrink until it hits other operators. Therefore, the symmetry charge $Q_\xi$ acting on a local operator still

---

[2]Here we use the Euclidean signature. For Lorentzian CFT, the symmetry group is $SO(d,2)$

yields another local operator by conformal transformation in spite of its non-local appearance.[3] As a familiar example, in Lorentzian signature we have the ordinary Hamiltonian $H = -\int d^{d-1}x T_{00} = \int d^{d-1}x T^{tt}$ associated to the time translation $\xi^0 = $ const, here the surface $\Sigma$ is chosen to be a spatial slice in the Cartesian coordinate. In general, we can choose another conformal Killing vector $\xi^\mu$ and regard the associated symmetry charge $Q_\xi$ as a "deformed Hamiltonian", and use it to construct an evolution operator $\exp(-itQ_\xi)$ that acts on states on $\Sigma$. Such evolution has a nice property that its action on local operators can be fully characterized by a conformal transformation generated by $\xi^\mu$.

Now for $d = 2$, in addition to the "global" conformal group $SO(3,1) \simeq SL(2,\mathbb{C})$ that has been used to generate the "$SL_2$ deformed Hamiltonian" as discussed in Ref. [47–55], we can also exploit the enhanced symmetry, i.e. the infinite dimensional Virasoro symmetry, to construct more general deformed Hamiltonians. More explicitly, at $d = 2$ the conformal Killing equation

$$\partial_\mu \xi_\nu + \partial_\nu \xi_\mu - \frac{2}{d}\eta_{\mu\nu}\partial^\mu \xi_\mu = 0 \tag{3}$$

reduces to the equations

$$\partial_{\bar{z}}\xi^z = 0, \quad \partial_z \xi^{\bar{z}} = 0, \tag{4}$$

where we have adopted the complex coordinates

$$z = x^0 + ix^1, \quad \bar{z} = x^0 - ix^1, \quad \partial_z = \frac{1}{2}\left(\partial_0 - i\partial_1\right), \quad \partial_{\bar{z}} = \frac{1}{2}\left(\partial_0 + i\partial_1\right). \tag{5}$$

Vector $\xi^z = \xi^0 + i\xi^1$, $\xi^{\bar{z}} = \xi^0 - i\xi^1$ follows the same rule as $z = x^z$ and $\bar{z} = x^{\bar{z}}$. The conformal Killing equation (4) indicates that at $d = 2$, we have an infinite set of solution $\xi^z(z,\bar{z}) = \xi(z)$, $\xi^{\bar{z}}(z,\bar{z}) = \bar{\xi}(\bar{z})$, where $\xi(z)$ and $\bar{\xi}(\bar{z})$ are two arbitrary independent holomorphic and anti-holomorphic functions. With $\xi$ and $\bar{\xi}$, we can express the charges associated to the conformal symmetry as follows (let $\Sigma$ be the spatial slice at $x^0 = 0$ momentarily and parametrized by $x = x^1$ for simplicity)

$$Q_\xi = -\int dx(\xi^0 T_{00} + \xi^1 T_{01}) = -\int dx \left(\xi T_{zz} + \bar{\xi}T_{\bar{z}\bar{z}}\right). \tag{6}$$

In the second step, we have used the tracelssness of the energy-momentum tensor $T^\mu_\mu = 0$ to eliminate the $T_{z\bar{z}}$ and $T_{\bar{z}z}$ component. And the conservation law $\partial^\mu T_{\mu\nu} = 0$ implies that $\partial_{\bar{z}}T_{zz} = \partial_z T_{\bar{z}\bar{z}} = 0$, namely $T_{zz}$ is chiral and $T_{\bar{z}\bar{z}}$ anti-chiral. It is customary to use the following notation

$$T(z) = -2\pi T_{zz}, \qquad \overline{T}(\bar{z}) = -2\pi T_{\bar{z}\bar{z}}. \tag{7}$$

Therefore, the deformed Hamiltonian on the real line has the following form

$$Q_\xi = \int \frac{dx}{2\pi}\left(\xi(z)T(z) + \bar{\xi}(\bar{z})\overline{T}(\bar{z})\right)|_{z=-\bar{z}=ix}, \tag{8}$$

where the integration is over section $z = -\bar{z} = ix$. This is just (1) but defined on the infinite real line. In the following, we will first derive the result on the real line and translate to the

---

[3]Note that under this definition, the symmetry charge here does not have to commute with the "Hamiltonian", which itself is a charge associated with a time-like Killing vector in Lorentzian signature. See e.g. Ref. [46] for an exposition on this subject.

cylinder through conformal transformation. For a general surface $\Sigma$ parametrized by $(z, \overline{z})$, we have

$$Q_\xi(\Sigma) = \oint_\Sigma \frac{1}{2\pi i} \left( dz\, \xi(z) T(z) - d\overline{z}\, \overline{\xi}(z) \overline{T}(z) \right). \tag{9}$$

## 2.2 Operator evolution on complex plane

Now for a chiral primary field $O(z)$ at surface $\Sigma$ with conformal weight $h$, we can derive the operator evolution $O_t(z) := e^{itQ_\xi(\Sigma)} O(z) e^{-itQ_\xi(\Sigma)}$ via Heisenberg equation w.r.t. the deformed Hamiltonian, i.e. $\frac{d}{dt} O_t(z) = i[Q_\xi(\Sigma), O_t(z)]$. At $t = 0$, the commutator can be obtained via the OPE of $T$ and $O$

$$[Q_\xi(\Sigma), O(z)] = \oint_C \frac{dw}{2\pi i} \xi(w) T(w) O(z) = (h\xi'(z) O(z) + \xi(z) \partial O(z)) \tag{10}$$

where contour $C$ circles $z$ counter-clockwisely. The evolution can be recasted as a coordinate transformation generated by $\xi$. More explicitly, for an infinitesimal $t = \varepsilon$, we have

$$O_\varepsilon(z) = \left( \frac{\partial z_\varepsilon}{\partial z} \right)^h O(z_\varepsilon), \quad \text{with} \quad z_\varepsilon = z + i\varepsilon\xi(z), \quad \text{at} \quad \varepsilon \to 0. \tag{11}$$

Thus, the operator evolution at finite $t$ is

$$O_t(z) = \left( \frac{\partial z_t}{\partial z} \right)^h O(z_t), \quad \text{with} \quad \frac{dz_t}{dt} = i\xi(z_t). \tag{12}$$

For the chiral energy-momentum tensor $T(z)$, we will have a central charge term for the conformal transformation $z \xrightarrow{\xi}_t z_t$

$$T_t(z) = \left( \frac{\partial z_t}{\partial z} \right)^2 T(z_t) + \frac{c}{12} \text{Sch}(z_t, z) \tag{13}$$

where $c$ is the central charge and $\text{Sch}(y, x)$ stands for the Schwarizan derivative

$$\text{Sch}(y, x) = \frac{y'''(x)}{y'(x)} - \frac{3}{2} \left( \frac{y''(x)}{y'(x)} \right)^2. \tag{14}$$

Similarly, the operator evolution of the anti-chiral primaries and energy-momentum tensor are determined by the vector field $\overline{\xi}(\overline{z})$.

In summary, we confirmed that the operator evolution driven by $\exp(-itQ_\xi(\Sigma))$, where $Q_\xi(\Sigma)$ is a deformed Hamiltonian (9), is given by a conformal transformation $(z, \overline{z}) \to (z_t, \overline{z}_t)$ generated by the flow of vector fields $(\xi(z), \overline{\xi}(\overline{z}))$

$$\boxed{\frac{dz_t}{dt} = i\xi(z_t), \qquad \frac{d\overline{z}_t}{dt} = i\overline{\xi}(\overline{z}_t).} \tag{15}$$

In particular, the above discussions are performed at Euclidean signature, and $t$ is regarded as a parameter, not confused with "time" $x^0$.

## 2.3 Deformation as velocity profile

Now let us move to the Lorentzian signature and check the meaning and implication of the above transformation law in real time.

In Lorentzian signature, we wick rotate $x^0 \to ix^0$ and therefore

$$z = i(x^0 + x^1), \quad \overline{z} = i(x^0 - x^1). \tag{16}$$

For simplicity, we put the surface $\Sigma$ at time $x^0 = 0$ in this discussion and focus on the chiral part first. A chiral operator $O(z)$ (i.e. obeying $\partial_{\overline{z}} O = 0$) travels along the lightcone $x^1 = x^0 + x_{\text{initial}}$ when we increase time $x^0$. Now, when applying $U = e^{-itH_\xi}$ to the chiral operator $O_t(z) = U^{-1}O(z)U$, we can have two interpretations for the conformal transformation $z = i(x^0 + x^1) \to z_t = i(x_t^0 + x_t^1)$, as follows:

1. It can be understood as a shift on the time coordinate $x^0$ while keep the spatial coordinate $x^1$ fixed, namely

$$\frac{dx_t^0}{dt} = \xi_x(x_t^0), \quad \text{with} \quad \xi_{x^1}(x^0) := \xi(z)|_{z=i(x^0+x^1)}. \tag{17}$$

   Here the subscript $x^1$ denotes the spatial location of the operator $O(z)$ at time $x^0 = 0$. In this interpretation, we regard the vector $\xi(z)$ as governing the uneven time flow for chiral operators at fixed location. Similarly, the vector $\overline{\xi}(\overline{z})$ governs the time flow for anti-chiral operators.

2. It can also be understood as a shift on the spatial coordinate $x^1$ with fixed time $x^0 = 0$, namely

$$\frac{dx_t^1}{dt} = v(x_t^1), \quad \text{with} \quad v(x^1) := \xi(z)|_{z=ix^1}. \tag{18}$$

   In this interpretation, we stay in the same time slice, and regard $\xi$ as governing the non-uniform traveling "velocity" of the chiral operator in the spatial direction. For the anti-chiral operator with $\overline{z} = i(x^0 - x^1)$, we have

$$\frac{dx_t^1}{dt} = -\overline{v}(x_t^1), \quad \text{with} \quad \overline{v}(x^1) := \overline{\xi}(\overline{z})|_{\overline{z}=-ix^1}. \tag{19}$$

In this paper, we will take the second interpretation, and rename the symmetry charge $Q_\xi$ (8) as the deformed Hamiltonian $H_v$

$$H_v = \int \frac{dx}{2\pi} \left( v(x)T(x) + \overline{v}(x)\overline{T}(x) \right) \tag{20}$$

with the smooth velocity functions $(v, \overline{v})$ identified with the conformal Killing vectors $(\xi, \overline{\xi})$ at the spatial slice. For the infinite line discussed above, we have $(v(x), \overline{v}(x)) = (\xi(z)|_{z=ix}, \overline{\xi}(\overline{z})|_{\overline{z}=-ix})$, representing the right and left moving velocities for chiral and anti-chiral operators.

## 2.4 Operator evolution on cylinder

In the main text, we will consider a finite system of length $L$ with periodic boundary condition, which can be realized by the unit circle on the complex plane

$$z = \exp\left(\frac{2\pi}{L}w\right), \quad \overline{z} = \exp\left(\frac{2\pi}{L}\overline{w}\right); \qquad w = \tau + ix, \quad \overline{w} = \tau - ix \tag{21}$$

namely we let the surface $\Sigma$ locate at $\tau = 0$ and be parametrized by $x \in [0, L]$. Now the time translation along $\tau$ is related to the dilation in radial direction, i.e. the corresponding conformal Killing vectors are $\xi(z) = z$, $\overline{\xi}(\overline{z}) = \overline{z}$, and the associated dilation operator derived from (9) has the following form

$$D = \frac{L}{2\pi} \cdot \int_0^L \frac{dx}{2\pi} \left(T(w) + \overline{T}(\overline{w})\right) + \frac{c + \overline{c}}{24} \tag{22}$$

where we have used the transformation rule for the chiral energy-momentum tensor

$$T(w) = \left(\frac{\partial z}{\partial w}\right)^2 T(z) + \frac{c}{12} \mathrm{Sch}\,(z, w) = \left(\frac{2\pi}{L}\right)^2 \left[e^{\frac{4\pi}{L}w}T(z) - \frac{c}{24}\right] \tag{23}$$

and similarly for the anti-chiral part. It is customary to identify the integral in (22) as the (ordinary) Hamiltonian on the circle, namely

$$H = \int_0^L \frac{dx}{2\pi}(T(w) + \overline{T}(\overline{w}))|_{w=-\overline{w}=ix}. \tag{24}$$

Now let us consider a deformation generated by $(\xi, \overline{\xi})$ that is holomorphic and anti-holomorphic on cylinder parametrized by $\tau \in (-\infty, \infty)$ and $x \in [0, L]$. In the $(z, \overline{z})$ coordinate, the cylinder regime corresponds to the complex plane with origin removed, i.e. $\mathbb{C} - \{0\}$. Therefore, we can write $\xi$ and $\overline{\xi}$ in Laurent series

$$\xi(z) = z \sum_{n=-\infty}^{+\infty} \widetilde{v}_n z^n, \qquad \overline{\xi}(\overline{z}) = \overline{z} \sum_{n=-\infty}^{+\infty} \widetilde{\overline{v}}_n \overline{z}^n \tag{25}$$

here we factor out a $z$ and $\overline{z}$ for later convenience in notation.[4] Inserting into (9), we obtain the corresponding topological surface operator evaluated at the unit circle as follows

$$Q_\xi = \frac{L}{2\pi} \int_0^L \frac{dx}{2\pi} \left(v(x)T(x) + \overline{v}(x)\overline{T}(x)\right) + \frac{v_0 c + \overline{v}_0 \overline{c}}{24} \tag{26}$$

where $v(x)$ and $\overline{v}(x)$ are the Fourier transform of $\{v_n\}$ and $\{\overline{v}_n\}$

$$v(x) = \sum_{n=-\infty}^{+\infty} \widetilde{v}_n e^{i\frac{2\pi n}{L}x}, \qquad \overline{v}(x) = \sum_{n=-\infty}^{+\infty} \widetilde{\overline{v}}_n e^{-i\frac{2\pi n}{L}x}. \tag{27}$$

---

[4]The $SL_2$ deformed Hamiltonian studied in Ref. [38, 40, 43] corresponds to the case with $v_{-q,0,q}$ (and $\overline{v}_{-q,0,q}$) only, i.e. conformal Killing vector $\xi(z)$ (and $\overline{\xi}(\overline{z})$) is a quadratic polynomial of $z$ (and $\overline{z}$).

Similarly, we will single out the integral part in (26) as our deformed Hamiltonian on the circle

$$H_v = \int_0^L \frac{dx}{2\pi} \left( v(x)T(x) + \overline{v}(x)\overline{T}(x) \right) \tag{28}$$

Note it has the similar form as the deformed Hamiltonian on infinite line (20) except the integral domain.

We can also derive the transformation law for the coordinates of a chiral operator $O(w)$ following (15). Let $O_t = e^{itH_v}Oe^{-itH_v} = e^{it\frac{2\pi}{L}Q_\xi}Oe^{-it\frac{2\pi}{L}Q_\xi}$, then we have

$$\frac{dz_t}{dt} = i\frac{2\pi}{L}\xi(z_t) \Rightarrow \frac{d}{dt}e^{\frac{2\pi}{L}w_t} = i\frac{2\pi}{L}e^{\frac{2\pi}{L}w_t}\underbrace{\sum_{n=-\infty}^{+\infty} v_n e^{\frac{2\pi}{L}nw_t}}_{v(-iw_t)} \tag{29}$$

$$\Rightarrow \frac{dw_t}{dt} = iv(-iw_t). $$

Further restrict to the $\tau = 0$ slice (i.e. $w = ix$), we have

$$\frac{dx_t}{dt} = v(x_t) \tag{30}$$

For the anti-chiral operator $O(\overline{w})$, we will get $\frac{dx_t}{dt} = -\overline{v}(x_t)$. Note these two equations are of the same forms as the trajectories (18) (19) for operators on the infinite line.

## 2.5 Entanglement and energy

With the above formalism, let us summarize the recipe to determine the operator evolution and apply it to the entanglement and energy calculation on a finite circle with length $L$

1. For a given deformed Hamiltonian (28) characterized by $(v(x), \overline{v}(x))$, we can determine the corresponding conformal transformations $(w, \overline{w}) \to (w_t, \overline{w}_t)$ for holomorphic and anti-holomorphic parts as follows

$$\frac{dw_t}{dt} = iv(-iw_t), \qquad \frac{d\overline{w}_t}{dt} = i\overline{v}(i\overline{w}_t). \tag{31}$$

2. A primary operator $O(w, \overline{w})$ of weight $(h, \overline{h})$ transform as follows

$$O_t(w, \overline{w}) = \left(\frac{\partial w_t}{\partial w}\right)^h \left(\frac{\partial \overline{w}_t}{\partial \overline{w}}\right)^{\overline{h}} O(w_t, \overline{w}_t), \tag{32}$$

where the $O_t(w, \overline{w}) = e^{iH_vt}O(w, \overline{w})e^{-iH_vt}$ is the evolved operator. And the energy-momentum tensor transform as

$$T_t(w) = \left(\frac{\partial w_t}{\partial w}\right)^2 T(w_t) + \frac{c}{12}\,\text{Sch}(w_t, w),$$

$$\overline{T}_t(\overline{w}) = \left(\frac{\partial \overline{w}_t}{\partial \overline{w}}\right)^2 T(\overline{w}(t)) + \frac{c}{12}\,\text{Sch}(\overline{w}_t, \overline{w}). \tag{33}$$

In the applications to the Floquet system that will be discussed in the following section, we are interested in measuring the equal time two point function of primaries (e.g. twist operators for the entanglement entropy calculation) and one point function of energy-momentum tensor on evolved vacuum $|\psi\rangle = e^{-it_1 H_{v_1}} e^{-it_2 H_{v_2}} ... e^{-it_n H_{v_n}} |\text{vac}\rangle$

$$
\begin{aligned}
\langle O(x,n)O(y,n)\rangle &:= \langle e^{it_n H_{v_n}} ... e^{it_1 H_{v_1}} O(x)O(y) e^{-it_1 H_{v_1}} .. e^{-it_n H_{v_n}}\rangle, \\
\langle T(x,n)\rangle &:= \langle e^{it_n H_{v_n}} ... e^{it_1 H_{v_1}} T(x) e^{-it_1 H_{v_1}} .. e^{-it_n H_{v_n}}\rangle, \\
\langle \overline{T}(x,n)\rangle &:= \langle e^{it_n H_{v_n}} ... e^{it_1 H_{v_1}} \overline{T}(x) e^{-it_1 H_{v_1}} .. e^{-it_n H_{v_n}}\rangle
\end{aligned}
\tag{34}
$$

where we have labelled the operators with their spatial coordinate $x$ only as they are defined on $\tau = 0$ slice, i.e. $w = -\overline{w} = ix$. Therefore, it will be convenient (and intuitive) to express (34) in terms of their trajectories on the spatial direction as discussed in section 2.3, namely we introduce (formally) two spatial coordinates $(x_t, \overline{x}_t) = (-iw_t, i\overline{w}_t)$ representing the chiral and anti-chiral parts of the operator which obey the flow equation

$$
\boxed{\frac{dx_t}{dt} = v(x_t), \qquad \frac{d\overline{x}_t}{dt} = -v(\overline{x}_t).}
\tag{35}
$$

For a consecutive evolution $e^{-it_1 H_{v_1}} e^{-it_2 H_{v_2}} ... e^{-it_n H_{v_n}}$ of $n$ steps, we obtain $n$ end points $(x_1, x_2, ..., x_n)$ as smooth functions of initial position $x$

$$
x \xrightarrow[t_1]{v_1} x_1 \xrightarrow[t_2]{v_2} x_2 \xrightarrow[t_3]{v_3} ... \xrightarrow[t_n]{v_n} x_n
\tag{36}
$$

and similarly for $(\overline{x}_1, \overline{x}_2, ..., \overline{x}_n)$ as functions of $\overline{x}$.

With this, we can express the correlation function in (34) as follows

$$
\langle O(x,n)O(y,n)\rangle = \frac{\left(\frac{\partial x_n}{\partial x} \frac{\partial y_n}{\partial y}\right)^h \left(\frac{\partial \overline{x}_n}{\partial \overline{x}} \frac{\partial \overline{y}_n}{\partial \overline{y}}\right)^{\overline{h}}}{\left[\frac{L}{\pi} \sin\left(\frac{\pi}{L}(x_n - y_n)\right)\right]^{2h} \left[\frac{L}{\pi} \sin\left(\frac{\pi}{L}(\overline{x}_n - \overline{y}_n)\right)\right]^{2\overline{h}}}
\tag{37}
$$

from which we can deduce the entanglement (von Neumann) entropy for interval $A = [x, y]$ following [56–58]

$$
\boxed{S_A(n) = \frac{c}{12} \log \frac{\left[\frac{L^2}{\pi^2 \varepsilon^2} \sin\left(\frac{\pi}{L}(x_n - y_n)\right) \sin\left(\frac{\pi}{L}(\overline{x}_n - \overline{y}_n)\right)\right]^2}{\left(\frac{\partial x_n}{\partial x} \frac{\partial y_n}{\partial y} \frac{\partial \overline{x}_n}{\partial \overline{x}} \frac{\partial \overline{y}_n}{\partial \overline{y}}\right)}}
\tag{38}
$$

where the argument $n$ denotes the number of step of driving, not confused with $n$-th Renyi entropy. Here and in the following, we assume a non-chiral CFT with $c = \overline{c}$. The above formula is subject to a constant ambiguity from the UV cutoff, here denoted by $\varepsilon$.

Next, the expectation value of energy-momentum tensor[5]

$$
\boxed{
\begin{aligned}
\langle T(x,n)\rangle &= \left(\frac{\partial x_n}{\partial x}\right)^2 \langle T(x_n)\rangle - \frac{c}{12} \text{Sch}(x_n, x), \\
\langle \overline{T}(\overline{x},n)\rangle &= \left(\frac{\partial \overline{x}_n}{\partial \overline{x}}\right)^2 \langle \overline{T}(\overline{x}_n)\rangle - \frac{c}{12} \text{Sch}(\overline{x}_n, \overline{x}).
\end{aligned}
}
\tag{41}
$$

---

[5]If we insert the vacuum expectation value of energy-momentum tensor on circle $\langle T(x)\rangle = -\frac{c\pi^2}{6L^2}$ and use the

Note the sign change in front of the Schwarzian term as a consequence of the coordinate transform $\partial_w f(w) = -i\partial_x f(w)$ and $\partial_{\overline{w}} f(\overline{w}) = i\partial_x f(\overline{w})$ when acting on holomorphic and anti-holomorphic functions.[6]

# 3 Floquet CFT with general deformation

The formalism in the last section explains how the evolution of physical observable, such as energy and entanglement entropy, is determined by the smooth deformation $(v(x), \overline{v}(x))$ in Hamiltonian (28)

$$H_v = \int_0^L \frac{dx}{2\pi} \left( v(x)T(x) + \overline{v}(x)\overline{T}(x) \right) \tag{42}$$

For a consecutive evolution $U = e^{-it_n H_{v_n}}...e^{-it_2 H_{v_2}} e^{-it_1 H_{v_1}}$ of $n$ steps, the dynamics is encoded in the $n$ end points $(x_1, x_2, ..., x_n)$ of the flow

$$x \xrightarrow[t_1]{v_1} x_1 \xrightarrow[t_2]{v_2} x_2 \xrightarrow[t_3]{v_3} ... \xrightarrow[t_n]{v_n} x_n \tag{43}$$

as if a chiral operator $O(x)$ starting at $x$ is moved to $x_n$ after $n$-step of driving, and similarly for the anti-chiral part that is controlled by the vector field $\overline{v}(x)$.

In this framework, Floquet dynamics corresponds to the special case when we have periodicity built in the driving sequence $U$. In this paper, we are mostly interested in the long time asymptotics of the driven dynamics.

## 3.1 Two-step driving protocol

As a minimal setup, we consider the following two-step driving protocol:

$$\tag{44}$$

That is, the evolution

$$U = \underbrace{e^{-iT_0 H_0} e^{-iT_1 H_1}}_{\text{1 cycle}} ...e^{-iT_0 H_0} e^{-iT_1 H_1} = (e^{-iT_0 H_0} e^{-iT_1 H_1})^n \tag{45}$$

---

following identity

$$\mathrm{Sch}\left(e^{i\frac{2\pi x_n}{L}}, x\right) = \mathrm{Sch}(x_n, x) + \frac{1}{2}\left(\frac{2\pi}{L}\right)^2 \left(\frac{\partial x_n}{\partial x}\right)^2 \tag{39}$$

we can group the two terms in (41) together

$$\langle T(x, n) \rangle = -\frac{c}{12} \mathrm{Sch}\left(e^{i\frac{2\pi x_n}{L}}, x\right). \tag{40}$$

[6]For the scaling factor $\partial_x x_n$, there is no additional sign when transforming from $\partial_w w_n$ since $x_n = -iw_n$. However, the Schwarzian term (14) involves two un-cancelled derivatives and therefore leads to a sign flip in (41). Alternatively, one can derive (41) from commutation relation of the energy-momentum tensor and confirm the sign here.

consists of $n$ repeating cycles, with each cycle generated by two non-commuting Hamiltonians $H_1$ and $H_0$, for time duration $T_1$ and $T_0$ respectively.[7]

Now following the framework introduced in the last section, the operator evolution is fully characterized by the two step conformal transformation generated by the vector flow $(v_0, T_0)$, $(v_1, T_1)$

$$x \xrightarrow[T_0]{v_0} y \xrightarrow[T_1]{v_1} x_1 \xrightarrow[T_0]{v_0} y_1 \xrightarrow[T_1]{v_1} x_2 ... \xrightarrow[T_1]{v_0} x_n \tag{46}$$
$$\underbrace{\phantom{x \xrightarrow[T_0]{v_0} y \xrightarrow[T_1]{v_1} x_1}}_{1 \text{ cycle}}$$

where we denote the end point of middle step by $y_j$ and the end point of a cycle by $x_j$, obeying

$$T_0 = \int_{x_j}^{y_j} \frac{dx}{v_0(x)}, \qquad T_1 = \int_{y_j}^{x_{j+1}} \frac{dx}{v_1(x)} \quad \text{for} \quad j = 0, 1, ..., n-1. \tag{47}$$

For later convenience, let us denote the 1-cycle flow by a smooth orientation preserving map $f \in \mathrm{Diff}^+(S^1)$: $x_1 = f(x)$, then the flow of $n$-consecutive cycle is given by the $n$-th composition

$$x_n = f^n(x) := f \circ f \circ \cdots \circ f(x) \tag{48}$$

and similarly for the anti-chiral part $\bar{x}_n = \bar{f}^n(\bar{x})$.

## 3.2 Revisiting the $\mathrm{SL}_2$ deformation and fixed points

In the $\mathrm{SL}_2$ deformed Floquet CFT [38–44], we modulate the Hamiltonian by a single wavelength, namely we let

$$v(x) = a + b \cdot \sin \frac{2\pi q x}{L} + c \cdot \cos \frac{2\pi q x}{L}, \quad a, b, c \in \mathbb{R}, \quad q \in \mathbb{Z}. \tag{49}$$

and similarly for $\bar{v}(x)$ in the deformed Hamiltonian (42). With this choice, the deformed Hamiltonian can be expressed as a linear superposition of two copies of $\mathfrak{sl}_2$ (chiral and anti-chiral), i.e. $\{L_0, L_q, L_{-q}\} \oplus \{L_0, \bar{L}_{\bar{q}}, \bar{L}_{-\bar{q}}\}$, here $\bar{q}$ could be a different integer from $q$.

For an illustration, let us take $v(x) = 2\sin^2(\frac{\pi q x}{L})$ (with $q > 1$), and $\bar{v}(x) = 0$ in the deformed Hamiltonian $H_1$

$$H_1 = \int_0^L \frac{dx}{2\pi} \left( v(x)T(x) + \bar{v}(x)\bar{T}(x) \right) \tag{50}$$

and $H_0$ is chosen to be the homogeneous one with $v(x) = \bar{v}(x) = 1$. Using (47), we find

$$T_1 = \frac{1}{\frac{2\pi q}{L} \tan\left(\frac{\pi q x_j}{L}\right)} - \frac{1}{\frac{2\pi q}{L} \tan\left(\frac{\pi q y_j}{L}\right)}, \quad T_0 = x_{j+1} - y_j. \tag{51}$$

Eliminating $y_j$, we find the map $x_{j+1} = f(x_j)$ is determined by the following equation

$$\cot \frac{q\pi(x_j + T_0)}{L} - \cot \frac{q\pi x_{j+1}}{L} = \frac{2\pi q T_1}{L}. \tag{52}$$

---

[7]This minimal setup can be straightforwardly generalized to the case with multiple steps of driving within a cycle, or quasi-periodic driving, as discussed in Ref. [43] where the Hamiltonian is spatially modulated by a single wavelength. In this paper, we keep the driving protocol simple but allow the deformation general.

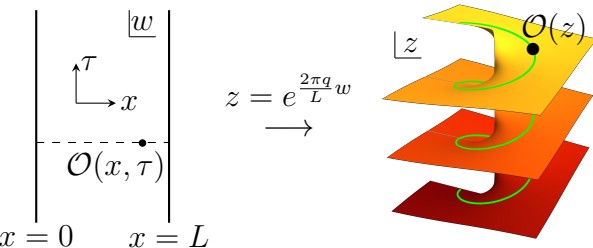

Figure 3: For an operator $O(w)$ with $w = \tau + ix$ on a finite system of length $L$ with periodic boundary condition, we can map it to an operator on $q$-sheet Riemann surface via $z = e^{\frac{2\pi q}{L}w}$, where the operator evolution by $\text{SL}_2$ deformed Hamiltonian (49) can be recast into a Möbius transformation.

Using complex variable $z = e^{\frac{2\pi q}{L}w}$, $w = \tau + ix$, the above formula is equivalent to a Möbius transformation on the $q$-sheet Riemann surface (see Fig. 3)

$$z_{j+1} = \frac{\alpha z_j + \beta}{\beta^* z_j + \alpha^*}, \quad \text{with} \quad \alpha = (1 + \frac{i\pi T_1}{L})e^{\frac{i\pi T_0}{L}}, \quad \beta = -\frac{i\pi T_1}{L}e^{-\frac{i\pi T_0}{L}} \tag{53}$$

which has also been obtained in [38, 40, 43, 44] from a different perspective. That is to say, in the special case with $\text{SL}_2$ deformation, the flow equation $x_{j+1} = f(x_j)$ in $x$-frame corresponds to a Möbius transformation in $z$-frame.

In the aforementioned references, the phase diagrams of the $\text{SL}_2$ deformed Floquet CFT was obtained by analyzing the motion of points on the unit circle $|z| = 1$ under the Möbius transform (53) as follows

1. Non-heating phase: The points oscillate around the unit circle under the map $z_j \to z_{j+1}$. This corresponds to an elliptic Möbius transformation with $|\alpha + \alpha^*| < 2$ in (53). In this phase, both the entanglement entropy and the total energy oscillate in time.

2. Heating phase: There is a pair of stable-unstable[8] fixed point emergent in the map $z_j \to z_{j+1}$. Points on the circle move exponentially close to the stable fixed point under the Floquet driving, which corresponds to a hyperbolic Möbius transformation with $|\alpha + \alpha^*| > 2$ in (53). In this phase, the entanglement entropy grows linearly in time, and the total energy of the system grows exponentially in time. In addition, the energy is mostly accumulated at the unstable fixed point (See, e.g., Fig. 1(c)).

3. Phase transition: The pair of stable-unstable fixed point merge to a single point with one side flow-in and the other side flow-out (will be called critical fixed point in this paper). The operator will be polynomially close to the stable fixed point as we increase the number of driving cycles, which corresponds to a parabolic Möbius transformation with $|\alpha + \alpha^*| = 2$ in (53). In this phase, the entanglement entropy grows logarithmically, and the total energy grows polynomially.

When we translate the above description in the $z$ frame back to the $x$ frame where the physical system is defined, we encounter a subtlety as we unfold the $q$-sheet Riemann surface

---

[8]characterized by whether the points in vicinity will flow towards or away respectively

by inverting the map $z = e^{\frac{2\pi q}{L}ix}$ (restricted to unit circle $\tau = 0$ slice already), due to the multivalueness of the inverse map $z \to (x, x + \frac{L}{q}, x + \frac{2L}{q}, ..., x + \frac{(q-1)L}{q})$. Therefore, a fixed point in the $z$ frame corresponds to $q$ fixed points in the $x$ frame, and there could be cases that a fixed point in the $x$-frame needs more than 1 cycle to return to itself, see Fig. 5 for an illustration.

In general, for an orientation preserving diffeomorphism $f$ with $q$ fixed point $\{x_{*,j}\}_{j=1,2...,q}$ on circle, let us denote its period by $p$, i.e.

$$\boxed{x_* = \underbrace{f \circ f \circ \cdots \circ f}_{p\,\text{times}}(x_*)} \tag{54}$$

where $p$ is the minimal integer that makes the equation hold. Then we know that $p$ is a divisor of $q$. This is because as we label the fixed point by 1 to $q$, the order has to be preserved under $f$ and therefore the permutation among them after each cycle is simply a shift by $s$, with $1 \leqslant s \leqslant q$ (shift is defined modulo $q$). Then the period $p = q/\gcd(q,s)$, where gcd stands for the greatest common divisor.[9] Analogous to the single period fixed point, for the period-$p$ fixed point $x_* = f^p(x_*)$, we can define the stable (denoted as $x_\bullet$), unstable (denoted as $x_\circ$) and critical fixed point (denoted as $x_c$) by checking the flow of the points in the vicinity of the fixed point under $f^p$ (see Fig. 4 for an illustration). More explicitly, we call $x_*$

1. a stable fixed point if $\frac{df^p(x)}{dx}|_{x=x_*} < 1$,

2. an unstable fixed point if $\frac{df^p(x)}{dx}|_{x=x_*} > 1$,

3. a critical fixed point if $\frac{df^p(x)}{dx}|_{x=x_*} = 1$, as it can be regarded as a degenerate case when a stable fixed point and an unstable fixed point merge together.

Note in our setting, $\frac{df^p(x)}{dx}|_{x=x_*} > 0$ since $f$ is orientation preserving. So far, we have only considered the chiral part. The discussion for the anti-chiral part simply follows. For convenience, we will only present the discussions for chiral part in the rest of the section (except the numerical simulation part where the lattice model naturally comes with two parts together). One can interpret this simplification as if we have only modified the chiral part of the Hamiltonian, i.e. only $v(x)$ is nonzero while $\bar{v}(x) = 0$ as we have assumed for the SL$_2$ case in the beginning of this section.

In the end, let us comment on the number of fixed points $q$: for a SL$_2$ deformed Floquet CFT generated by $L_0$ and $L_{\pm q}$, the total number of pairs of fixed point is given by the label of the Virasoro generator $L_q$. However, in a Floquet CFT with general deformations, as we will see in Sec. 3.4, both the total number $q$ and the periods $p$ of the fixed points may vary with driving parameter $(T_0, T_1)$ in the heating phase. That is, the internal structures of the heating phase with general deformations are much richer.

---

[9]For an explicit example showing this result, let us consider a the two-step driving in (44) with $H_0$ the homogeneous (undeformed) Hamiltonian and $H_1$ with $v(x) = 2\sin^2(\frac{\pi q x}{L})$, there are $q$ pairs of fixed point in heating phase, and the period of fixed points is determined by parameter $T_0$ (driving time of $T_0$): for $(s-1)/q < T_0/L < s/q$ where $1 \leqslant s \leqslant q$ is an integer, one can explicitly check that the period is given by $p = q/\gcd(q,s)$.

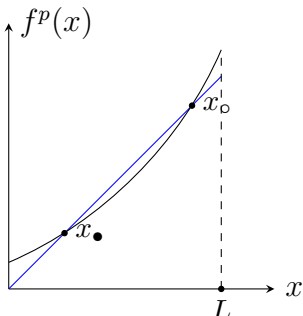

Figure 4: An illustration for fixed points of map $f^p$: the lower intersection $x_\bullet$ with diagonal is the stable fixed point with slope less than 1 and the higher intersection $x_\circ$ is the unstable fixed point with slope greater than 1.

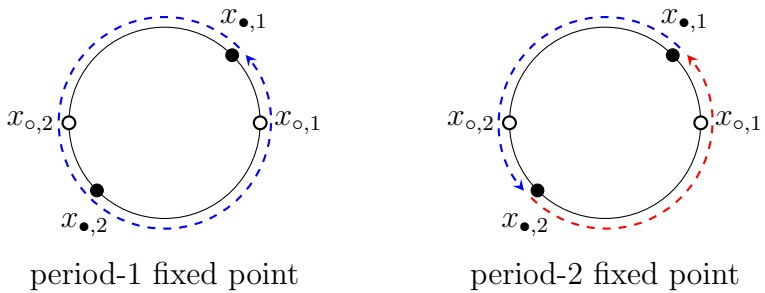

period-1 fixed point          period-2 fixed point

Figure 5: In the cartoon, we have $q = 2$, and one pair of fixed points in $z$ frame corresponds to two pairs in $x$, denoted by $x_{\bullet(\circ),j=1,2}$, where $\bullet(\circ)$ indicate that the fixed point is stable (unstable). Then there could be two scenarios that are indistinguishable in $z$ frame, but have different period in $x$ frame. In the first case shown above, $x_{\bullet(\circ),j}$ will flow back to itself after each driving cycle. In the second case, $x_{\bullet(\circ),1}$ will flow to $x_{\bullet(\circ),2}$ after one driving cycle, and flow back to itself after another driving cycle.

## 3.3   Two-step Floquet with general deformation

Unlike the $SL_2$ deformation, the underlying algebra for a general deformation $v(x)$ is the infinite dimensional Virasoro algebra, and therefore it is challenging to study the dynamics via an algebraic approach. Instead, we will directly work on the stroboscopic trajectory $x_{j+1} = f(x_j)$, which we classify into three classes:

1. No fixed point. In this case, the operators move around the circle with no accumulation point and therefore the correlation functions are oscillatory and in particular we expect that the entanglement entropy Eq. (38) will not grow in long time (large cycle number) asymptotics. Analogous to the $SL_2$ case [38, 40], we refer this class as non-heating phase.

2. Only stable and unstable fixed points are present. In this case, points on the circle can accumulate at (or diverge from) stable (unstable) fixed point $x_\bullet$ ($x_\circ$), which will correspond to the heating phase with growing entanglement following the discussion we will present momentarily.

3. Critical fixed points. This case can be identified with a phase transition between heating

and non-heating phase or between two heating phases with different number of fixed points, since the critical fixed point is a degenerate situation when a stable fixed point merges with an unstable one.

Now let us focus on the case 2, where only stable and unstable fixed point are present and comment on the critical fixed point in the end.

As discussed in the last section, in general we will encounter fixed point $x_*$ with period $p \geqslant 1$, namely $f^p(x_*) = x_*$ but not for smaller integers. Then the stability of the fixed point is determined by the derivative of $f^p$ at $x_*$. For instance, let us consider a point $x_* + \varepsilon$ near $x_*$ for small $\varepsilon$, then after map $f^p$, we have

$$\frac{f^p(x_* + \varepsilon) - f^p(x_*)}{x_* + \varepsilon - x_*} = \frac{df^p(x)}{dx}\bigg|_{x=x_*} \tag{55}$$

that is to say, for $\frac{df^p(x)}{dx}\big|_{x=x_*} < 1$, the distance to the fixed point is shrunken by a constant factor and we identify it as the stable fixed point, and rename $x_*$ by $x_\bullet$ in our notation. For later convenience, let us denote the derivative here by an exponent $\lambda(x_\bullet)$ as follows[10]

$$e^{-\lambda_\bullet} := \frac{df^p(x)}{dx}\bigg|_{x=x_\bullet}, \quad \lambda_\bullet > 0. \tag{57}$$

Similarly, for unstable fixed point $x_* = x_\circ$,

$$e^{\lambda_\circ} := \frac{df^p(x)}{dx}\bigg|_{x=x_\circ}, \quad \lambda_\circ > 0. \tag{58}$$

Thus, near the stable (unstable) fixed point, the deviation $\varepsilon_{\bullet(\circ),n} := (f^p)^n(x_{\bullet(\circ)} + \varepsilon) - x_{\bullet(\circ)}$ decreases (increases) exponentially

$$\varepsilon_{\bullet,n} = \varepsilon e^{-n\lambda_\bullet}, \qquad \varepsilon_{\circ,n} = \varepsilon e^{n\lambda_\circ} \tag{59}$$

which leads to the linear in $n$ behavior of entanglement growth in the heating phase that will be discussed in a moment. It is noted that $\lambda_\bullet$ and $\lambda_\circ$ are also known as the Lyapunov exponents of the map $f^p$.

Let us make a final comment on fixed points connected through the map $f$. For instance, let $x_\bullet$ and $y_\bullet$ be two period-$p$ fixed points satisfying $y_\bullet = f(x_\bullet)$. Then we take a point $x = x_\bullet + \delta x$ close enough to the fixed point $x_\bullet$ and apply $f^{p+1}$ to it. From one hand, we have $f(f^p(x)) = f(x_\bullet + \delta x e^{-\lambda_\bullet(x_\bullet)}) = y_\bullet + f'(x_\bullet)\delta x e^{-\lambda_\bullet(x_\bullet)}$, where we have used the condition that $x_\bullet$ is a stable fixed point. On the other hand, we can first map it to $y = f(x)$ and take $f^p$, i.e. $f^p(f(x)) = f^p(y_\bullet + f'(x_\bullet)\delta x) = y_\bullet + e^{-\lambda_\bullet(y_\bullet)}f'(x_\bullet)\delta x$. The consistency between the two results requires that all the fixed point in the same orbit of map $f$ must have the same exponent.

---

[10]One can also assume an effective Hamiltonian $H_{\text{eff}}$ with smooth deformation $v_{\text{eff}}$ and $\bar{v}_{\text{eff}}$ that reproduces the $p$ cycle driving, i.e. $e^{-ip(T_0+T_1)H_{\text{eff}}} = (e^{-iH_0T_0}e^{-iH_1T_1})^p$. Then the fixed points is related to the zeros of the deformation $v_{\text{eff}}(x_*) = 0$, and the slope is related to the exponents $\lambda_{\bullet(\circ)}$ as follows

$$\frac{df^p(x)}{dx}\bigg|_{x=x_*} = e^{v'_{\text{eff}}(x_*)p(T_0+T_1)} \implies \lambda_\bullet = -v'_{\text{eff}}(x_\bullet)p(T_0+T_1), \quad \lambda_\circ = v'_{\text{eff}}(x_\circ)p(T_0+T_1) \tag{56}$$

This also inspires a discussion on the quantum quench as shown in appendix A.

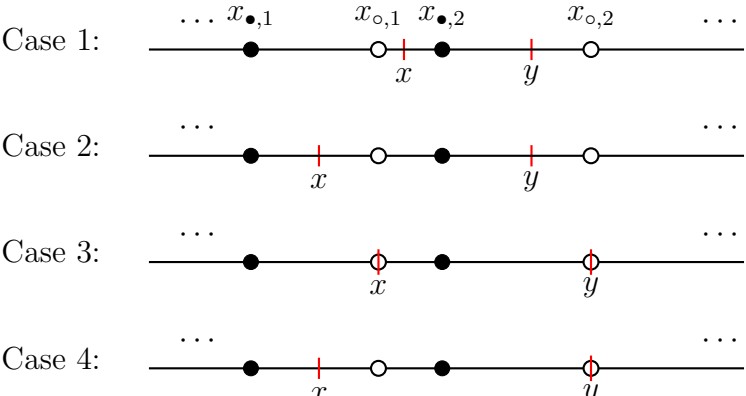

Figure 6: Different configurations of the locations of two operators (or the entanglement cuts $x$ and $y$) with respect to the unstable ($\circ$) fixed points in the stroboscopic trajectories of operator evolution.

### 3.3.1 Entanglement entropy

For entanglement entropy of a cut $A = [x, y]$ in the long time limit, we consider the following configurations (see Fig. 6)

1. There is no unstable fixed point of operator evolution in $[x, y]$. In this case, operator $O(x)$ and $O(y)$ will flow towards the same stable fixed point. In the long time limit, the two-point function will approach a finite and stable value and so does the entanglement entropy.

2. There is at least one (but not all) of unstable fixed points in $[x, y]$, and $x$ and $y$ do not coincide with any other unstable fixed points. In this case, $O(x)$ and $O(y)$ will flow to different stable fixed points $x_{\bullet,1}$ and $x_{\bullet,2}$ respectively. The chiral part of the two-point function after $np$ cycles ($n$ times of $p$-period map) is given by

$$\langle O(x, np) O(y, np) \rangle \simeq \left( e^{-(\lambda_{\bullet,1} + \lambda_{\bullet,2})n} \right)^h \left( \frac{L}{\pi} \sin \frac{\pi(x_{\bullet,1} - x_{\bullet,2})}{L} \right)^{-2h} \tag{60}$$

where $\lambda_{\bullet,1}, \lambda_{\bullet,2} > 0$ characterize the rate of operators approaching the stable fixed points. In turn, the entanglement entropy grows linearly in $n$ as

$$S_A(np) - S_A(0) \simeq \frac{c}{12} \log \frac{\sin^2(\pi(x_{\bullet,1} - x_{\bullet,2})/L)}{\sin^2(\pi(x - y)/L)} + \frac{c}{12} \cdot (\lambda_{\bullet,1} + \lambda_{\bullet,2}) \cdot n. \tag{61}$$

Physically, the stable fixed point in operator evolution corresponds to the unstable fixed point in state evolution, and therefore serves as a source emitting EPR pairs. The configuration in case 2 allows the subregion A to capture increasing number of entangling EPR pairs as illustrated in Fig. 1.

3. $x$ and $y$ are located at two different unstable fixed points $x_{\circ,1}$ and $x_{\circ,2}$ respectively. In this case, $O(x)$ and $O(y)$ will stay at these two unstable fixed points, and the correlation

function will increase in $n$ as follows

$$\langle O(x, np)O(y, np)\rangle \simeq \left(e^{(\lambda_{\circ,1}+\lambda_{\circ,2})n}\right)^h \left(\frac{L}{\pi} \sin \frac{\pi(x_{\circ,1} - x_{\circ,2})}{L}\right)^{-2h} \tag{62}$$

where $\lambda_{\circ,1}, \lambda_{\circ,2} > 0$ characterize the rate of operators flowing away from the unstable fixed points. The entanglement entropy will decrease as follows

$$S_A(np) - S_A(0) \simeq \frac{c}{12} \log \frac{\sin^2(\pi(x_{\circ,1} - x_{\circ,2})/L)}{\sin^2(\pi(x-y)/L)} - \frac{c}{12} \cdot (\lambda_{\circ,1} + \lambda_{\circ,2}) \cdot n. \tag{63}$$

Physically, this linear decreasing is due to the fact that degrees of freedom that contribute to the entanglement flow to the entanglement cuts in time. The non-local EPR pairs now become local in real space and 'annihilate' with each other.

4. One of $x$ and $y$ is at unstable fixed point, and the other is not. Assuming that $O(x)$ stays at the unstable fixed point $x_{\circ,1}$, and $O(x)$ finally flows to the stable fixed point denoted by $x_{\bullet,2}$, then one has

$$\langle O(x, np)O(y, np)\rangle \simeq \left(e^{(\lambda_{\circ,1}-\lambda_{\bullet,2})n}\right)^h \left(\frac{L}{\pi} \sin \frac{\pi(x_{\circ,1} - x_{\bullet,2})}{L}\right)^{-2h} \tag{64}$$

where $\lambda_{\circ,1}, \lambda_{\bullet,2} > 0$. The correlation function depends on the competition between the stable and unstable fixed points. This competition effect can also be seen in the entanglement entropy evolution as

$$S_A(np) - S_A(0) \simeq \frac{c}{12} \log \frac{\sin^2(\pi(x_{\circ,1} - x_{\bullet,2})/L)}{\sin^2(\pi(x-y)/L)} + \frac{c}{12} \cdot (-\lambda_{\circ,1} + \lambda_{\bullet,2}) \cdot n. \tag{65}$$

Physically, it is because there is one source of EPR pairs, and at the same time one sink of EPR pairs. Notably, in a fine-tuned case $\lambda_{\circ,1} = \lambda_{\bullet,2}$, the entanglement will remain constant.

In addition, based on the same analysis in Ref. [40], one can find that each region $[x_{\circ,j}-\epsilon, x_{\circ,j}+\epsilon]$ centered at $x_{\circ,j}$ is mainly entangled with the nearby two regions $[x_{\circ,j-1} - \epsilon, x_{\circ,j-1} + \epsilon]$ and $[x_{\circ,j+1} - \epsilon, x_{\circ,j+1} + \epsilon]$, which contributes to the linear $n$ growth of entanglement entropy in (61). Finally, we hope to remark that the entanglement evolution discussed above is defined at the complete period, i.e. the total cycle number is $np$, where $p$ is the period of fixed points. If we look into the middle steps within a period, then one may observe fine structures in the entanglement evolution (See, e.g., Fig.1(c)).

At last, we discuss the critical fixed points, where we have $\frac{df^p(x)}{dx}|_{x=x_c} = 1$. Let us assume the next leading power after linear in the Taylor expansion is $a_m(x - x_c)^m$ with $m \geqslant 2$, namely we have the following expansion near critical point

$$f^p(x_c + \varepsilon) = x_c + \varepsilon + a_m \varepsilon^m + o(\varepsilon^m), \tag{66}$$

that is to say the deviation after a $p$-period driving, $\varepsilon_1 := f^p(x_c+\varepsilon) - x_c \approx \varepsilon + a_m\varepsilon^m$ is changed by a high power of $\varepsilon$. It is more convenient to express the change by

$$\frac{1}{\varepsilon_1^{m-1}} \approx \frac{1}{\varepsilon^{m-1}}(1 - (m-1)a_m\varepsilon^{m-1}) = \frac{1}{\varepsilon^{m-1}} - (m-1)a_m \tag{67}$$

as it can be used to find the deviation for larger $n$, $\varepsilon_n^{1-m} = \varepsilon^{1-m} - n(m-1)a_m$, where $\varepsilon_n := (f^p)^n(x_c + \varepsilon) - x_c$, here we focus on the stable fixed point with $a_m < 0$, i.e. case 2 in the configurations of Fig. 6. That means the distance from the fixed point changes as a fraction power of $n$ in stable fixed point, and therefore the entanglement entropy will grow logarithmically in $n$ as

$$S_A(np) - S_A(0) \simeq \frac{c}{12} \log \frac{\sin^2(\pi(x_{\bullet,1} - x_{\bullet,2})/L)}{\sin^2(\pi(x - y)/L)} + \frac{c}{12} \left( \frac{n_{\bullet,1}}{n_{\bullet,1} - 1} + \frac{n_{\bullet,2}}{n_{\bullet,2} - 1} \right) \log n. \qquad (68)$$

### 3.3.2 Energy-momentum density distribution

Finally, we consider how the fixed points manifest themselves in the evolution of the energy-momentum density. If the initial state is a primary state with conformal weight $(h, \bar{h})$, the chiral energy density after $np$ cycles is

$$
\begin{aligned}
\langle T(x, np) \rangle &= \left( \frac{\partial f^{np}(x)}{\partial x} \right)^2 \left( h - \frac{c}{24} \right) - \frac{c}{12} \operatorname{Sch}(f^{np}(x), x) \\
&= h \left( \frac{\partial f^{np}(x)}{\partial x} \right)^2 - \frac{c}{12} \operatorname{Sch}\left( \tan \frac{f^{np}(x)}{2}, x \right)
\end{aligned}
\qquad (69)
$$

where we have chosen $L = 2\pi$. On the second line, we explicitly separate the contribution from the conformal vacuum (second term) and the primary excitation (first term).

From our analysis on the entanglement entropy, we have seen that the entanglement entropy of a subsystem does not grow unless it encloses one unstable fixed point. Thus, it is natural to expect an energy peak to appear at unstable fixed points. This intuition is consistent with the first term. Namely, when $x = x_{\circ,j}$ is an unstable fixed point, then the first term will be exponentially growing at the late time

$$h \left( \frac{\partial f^{np}(x)}{\partial x} \right)^2 \approx h e^{2\lambda_{\circ,j} n}. \qquad (70)$$

And when $x$ is some generic point, the first term decays to be exponentially small in time. However, the second term might violate this simple picture and is actually the only term left for the evolution from the conformal vacuum state ($h = 0$). To reveal the subtlety here, we set $h = 0$ and calculate the change in energy density from one cycle to the next. At the unstable fixed point and late time, we have

$$\langle T(x_\circ, (n+1)p) \rangle - \langle T(x_\circ, np) \rangle \approx \frac{c}{24} e^{2\lambda_\circ n} \left( 1 - e^{2\lambda_\circ} - 2 \operatorname{Sch}\left( f^p(x), x \right) \Big|_{x=x_\circ} \right). \qquad (71)$$

Notice that all the time dependence comes from the prefactor $e^{2\lambda_\circ n}$, which is the same as identified above. Thus the magnitude is still exponentially growing. However, we do not have control on the sign of the term inside the bracket. When this term is positive and we have an exponentially growing peak, the Schwarzian part $\operatorname{Sch}\left( f^p(x), x \right)\big|_{x=x_\circ}$ can still be different for different unstable fixed points, which implies that the energy peaks do not have the same height. Moreover, when the term inside the bracket becomes negative, it leads to a negative

energy density that keeps decreasing unbounded from below[11] In the example discussed below, we do not encounter such a pathological behavior. However, we are not aware of a proof to exclude such thing to happen and leave this for a future study.

The growth rate of the (magnitude of) energy density is controlled by the unstable fixed point $\lambda_\circ$, while the growth rate of its entanglement entropy is related to the nearby stable fixed points $x_{\bullet,j\pm1}, \lambda_{\bullet,j\pm1}$. Thus, we generally do not expect a direct connection between the energy and entanglement growth for any single peak.[12] In our concrete example, the two growth rate turns out to be the same.

As a final remark, it is also interesting to consider the case that the initial state is a thermal ensemble at finite temperature $1/\beta$. In the high temperature limit $\beta \ll L$, one has $\langle T(x) \rangle = \mathrm{Tr}(e^{-\beta H} T(x)) = \frac{\pi^2 c}{6\beta^2}$. Then the chiral energy density becomes $\langle T(x, np) \rangle = (\frac{\partial f^{np}(x)}{\partial x})^2 \cdot \frac{\pi^2 c}{6\beta^2} - \frac{c}{12} \mathrm{Sch}(f^{np}(x), x)$. In the high temperature limit, the first term will dominate. One can observe that the peaks of $\langle T(x, np) \rangle$ are located at the unstable fixed points where $(\frac{\partial f^{np}(x)}{\partial x})^2$ grow exponentially in time.

## 3.4   Concrete example

In this section, we illustrate the features of generalized Flouqet CFTs with an explicit example.

It is a modification to the $SL_2$ deformed Floquet CFT reviewed above and thus reduces to the familiar $SL_2$ cases in certain limits. This helps demonstrate how different features are enriched and changed as we move away from the $SL_2$ limit. After we explain the setup, we first solve for the stroboscopic trajectories and fixed points, which determines the phase diagrams. We then focus on the spatial energy-entanglement pattern in different heating phases and finally close this section with a comparison between CFT results and lattice simulation.

### 3.4.1   Setup

The system is defined on a circle of length $L$. The two-step Floquet driving (44) is designed as follows. We choose $H_0$ to be the homogeneous Hamiltonian ($v(x) = 1$), $H_1$ to be the deformed one with the following deformation function

$$
v(x) = \begin{cases} 2\sin^2\left(\dfrac{\pi x}{L_A}\right), & 0 \leqslant x \leqslant L_A, \\ 2\sin^2\left(\dfrac{\pi(x - L_A)}{L_B}\right), & L_A < x \leqslant L, \end{cases} \qquad \tag{72}
$$

where $L_A + L_B = L$. It consists of two sine-square deformations with wavelengths $L_A$, $L_B$ respectively and glued at $x = L_A$ and $x = 0$ (or $L$).[13] We note that the two limits, namely

---

[11]The total energy is still positive. This is because our total energy is defined as the expectation valuf the Virasoro generator $L_0$, the spectrum of which is by itself bounded from below.

[12]In the $SL_2$ case, we have $\lambda_{\bullet,j} = \lambda_{\circ,j} = \lambda$, and thus all the energy peaks have the same height and the energy and entanglement growth are locally closely related.

[13]Note that so-defined $v(x)$ is not smooth for generic $L_A$, i.e. its second and higher order derivatives are not continuous. It will lead to discontinuity in physical observable, such as the entanglement and energy density. This issue can be remedied by smoothening out the deformation function near the gluing points. Such

$L_A = 0$ and $L_A = L_B = L/2$, correspond to the $\mathrm{SL}_2$ deformation with $q = 1$ and $q = 2$ respectively as reviewed in Sec. 3.2.

For generic $L_A \in (0, L)$, the deformation function involves infinite many Fourier components and thus the corresponding algebra becomes Virasoro algebra, enlarging the $\mathfrak{sl}_2$ discussed previously. As we will show momentarily, even small deviation from the $\mathrm{SL}_2$ limit enriches the phase diagram with structures beyond the $\mathrm{SL}_2$ prediction. However, they still falls into a unified description from the geometric viewpoint.

Our protocol is symmetric with respect to the exchange of $L_A$ and $L_B$, which allows us to only focus on $0 \leqslant L_A \leqslant L/2$. It is convenient to parametrize all the dimensionful quantities by the total system size, thus we introduce $x_m = L_A/L \in [0, 1/2]$ throughout the following discussion.

### 3.4.2 Stroboscopic trajectories and phase diagrams

This section concerns with the phase diagram of our example. In the $\mathrm{SL}_2$ limit ($x_m = 0$ or $x_m = 1/2$ in our example), the dynamics and hence the phase diagrams can be solved analytically, as shown by the first and last figure in Fig. 8. However, generic cases require numerics. Thanks to the geometric interpretation of the deformed Hamiltonian, the phase diagram can be determined by studying the trajectory of quasi-particles, as established in Sec. 3.3 and briefly reviewed below for reader's convenience.

Given a driving protocol, we can obtain the operator evolution using Eq. (35) and, most importantly, the stroboscopic trajectory of the coordinates for the operator, also dubbed as quasi-particle motion. The heating phase corresponds to the existence of fixed points, and the different heating phases are specified by the number and periodicity of fixed points. In the following, we first explain how we extract the two pieces of information and then show the phase diagrams.

Let us consider an enough number of points uniformly initialized on the spatial circle, and evolved by the conformal mapping that appears in the operator evolution formula. We keep track of their stroboscopic trajectories. Once there are fixed points, all trajectories (without fine tuned to the unstable ones) will flow towards the stable ones, which is a clear indicator of the number of fixed points. We can further determine the periodicity of those fixed points by tracing how each point moves in one cycle. Some typical examples are shown in Fig. 7, where we can observe two distinct features as tuning the driving parameters $T_0/L$ and $T_1/L$:

1. No fixed points. The trajectories keep winding around the circle, as seen in Fig. 7 (a). This corresponds to the non-heating phase.

2. Multiple fixed points. The trajectories converges to $q = 4, 3, 2$ fixed points, as shown in Fig. 7 (b), (c) and (d) respectively, which correspond to different heating phases. Here, the periodicity happens to equal the number of stable fixed points and Fig. 7 only shows the trajectories after every $q$ cycles. In addition, one can also identify unstable fixed points, on the two sides of which trajectories flow to different stable fixed points. In our example, we always find an equal number of stable and unstable fixed points. We also checked that these trajectories approach the fixed points exponentially close in time, which accounts

---

regularization does not change the physics and instead makes the problem less tractable. Therefore, we stick to the original deformation function and interpret the potential discontinuity in the results as artifact.

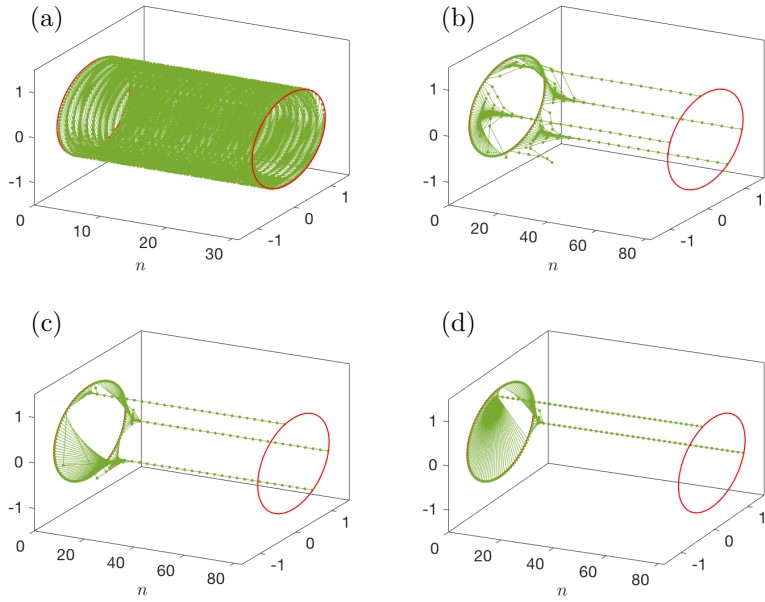

Figure 7: Stroboscopic trajectories (quasi-particle motion) for a fixed deformed Hamiltonian (72) and various driving parameters $T_0/L$, $T_1/L$. We choose $x_m = L_A/L = 0.3$ and $T_0/L = 0.1$ for all plots. We observed (a) no fixed points, (b) $q = 4$, (c) $q = 3$ and (d) $q = 2$ stable fixed points with $T_1/L = 0.05$, 0.2, 0.4, 0.8, respectively. In these examples, the number of stable fixed points equals their periodicity, i.e. $q = p$, and all trajectories are plotted for every $q$ driving cycles with $q = 4$, 3, 2, respectively. The unstable fixed points are identified with the bifurcation points of the trajectories.

for the exponential growth of the energy and linear growth of the entanglement entropy, shown in the following section.

Based on the knowledge of fixed points and the discussion in Sec. 3.3, one can obtain the phase diagrams, with typical ones shown in Fig. 1 and Fig. 8. In Fig. 8, we plot phase diagrams for different $x_m$ from 0 to 1/2. For $x_m = 0$ (the first figure), the driving reduces to the $q = 1$ SL$_2$ case. There is only one heating phase characterized by 1 pair of fixed points. Similarly, $x_m = 1/2$ (the last figure) corresponds to the $q = 2$ SL$_2$ case, and there are two heating phases. Both of them have two pairs of fixed points, while the one sitting in $0 < T_0/L < 1/2$ has period-2 fixed points and the other has period-1. However, such regularity gives way to the following richer patterns as $x_m$ deviates from the SL$_2$ limit to the Virasoro regime:

1. There are many different heating phases in the parameter space labeled by the number of (stable) fixed points $q$ and their periodicity $p$ with $p \leqslant q$. Since $q$ directly determines the energy-momentum distribution and is more physical, we choose $q$ to mark different heating phases in making the plots. For example, in Fig. 1 (a), there are four heating phases with 5 stable fixed points, all of which have period-5. As for the two heating phases with 2 stable fixed points, the left one has period-2 while the right one has period-1.

2. As the number of fixed points increase, the stroboscopic trajectories converge with lower

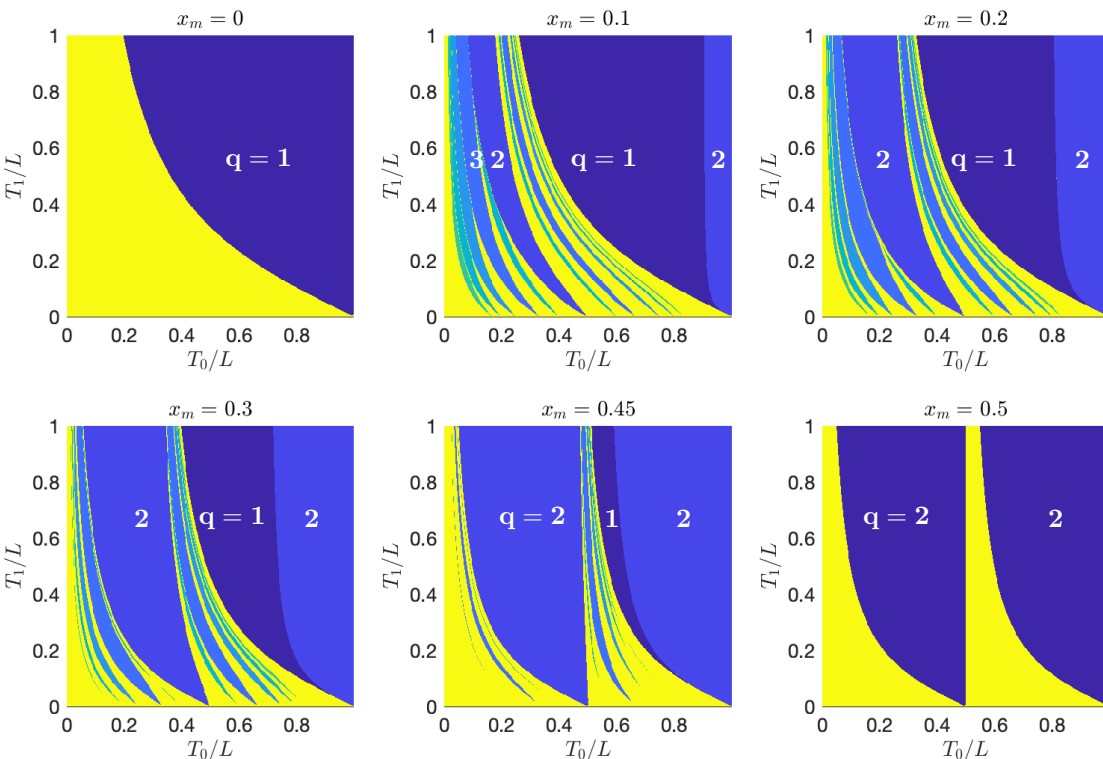

Figure 8: Phase diagram of a generalized Floquet CFT with different driving Hamiltonians in (72), where we choose (from left to right, and then top to bottom) $x_m = 0$, 0.1, 0.2, 0.3, 0.45, and 0.5. Also see Fig. 1(a) for a zoomed-in version of the $x_m = 0.3$ case. The regions in yellow (blue) correspond to the non-heating (heating) phases. Different heating phases are distinguished by the shades of color and the marks. Lighter blue regions have more fixed points (with higher periodicity). For $x_m = 0$ or 0.5, the setup reduces to a $SL_2$ deformed Floquet CFT and the phase diagram can be obtained analytically.

rate, which makes it numerically harder to identify the heating phase with large number of fixed points. For example, we identify heating phases only up to $q = 6$ in Fig. 1 (a).

3. If we look at heating phases with fixed points of period $p$, they always intersect the horizontal axis $T_1/L = 0$ at $T_0/L = r/p$ ($r$ and $p$ are co-prime). In the example we consider, the periodicity happens to be the same as the number of fixed points except for the right most $q = 2$ heating phase. Consequently, in Fig. 1 (a), the heating phase labeled with $q$ intersects the horizontal axis at $T_0/L = r/q$ and the right most $q = 2$ heating phase at $T_0/L = 1$. When $T_1/L = 0$ and $T_0 = r/p$, every point can be regarded as a period-$q$ fixed point. The above result seems to suggests that a small $T_1/L$ can lift the degeneracy and yields finitely many period-$p$ stable (and unstable) fixed points. [14] Such an observation also inspires us to conjecture that there can be infinite number of heating phases associated to all rational numbers $p/q$. The verification is left for future

_______________

[14]This phenomenon is similar to Arnol'd tongues in iterating circle maps [59]. We hope to study the possible relation to our phase diagram in the future.

study.

4. When $T_1/L$ increases, the $q = 2$ and $q = 1$ phases occupy larger region. This is because the dynamics in this regime is dominated by $H_1$, and the fixed point structure inherits that of $H_1$. In our example, $H_1$ has two fixed points at $x = 0, L_A$ which accounts for the large region of $q = 2$ phases. For certain values of $T_0/L$, one of the fixed points will be skipped. For example, one can consider $T_0/L = x_m + \epsilon$, then no matter where the initial position of the point is, it always flows to the fixed point at $x = 0$. This explains the appearance of the $q = 1$ phase and why it sits slightly on the right side of $T_0/L = x_m$ for large $T_1/L$.

5. The $q = 1$ phase with periodic boundary conditions has some non-generic behavior compared with the $q > 1$ phases. First, when $x_m = 0$ the initial state cannot be chosen as the ground state to give heating dynamics, as discussed in Ref. [39, 40]. Second, suppose we choose a general initial state such that the CFT is in a heating phase, then although there is a single (chiral) energy peak, the entanglement entropy does not grow even when the subsystem encloses the peak. This can be understood based on our quasi-particle picture that all the quasiparticles accumulate at the single chiral/anti-chiral peak and cannot contribute to entanglement. [15] This is different from the case of $q > 1$, where the quasiparticles can accumulate at different fixed points in real space, which contributes to a linearly growing entanglement entropy [39, 40].

### 3.4.3 Energy and entanglement evolution

In this section, we focus on the heating phases and calculate the evolution of chiral energy-momentum density $\langle T(x, n) \rangle$ and the entanglement entropy $S_A(n)$. The former exhibits growing peaks associated to the unstable fixed points of the conformal mapping, and the later reveals that they are the only resource of the entanglement generated by the Floquet driving. We choose the system size to be $L = 2\pi$, the initial state to be the ground state of $H_0$.

The evolution of the chiral energy-momentum density under generic deformed Hamiltonians follows (41). When the initial state is the ground state, we have

$$\langle T(x, n) \rangle = -\frac{c}{24} \left[ \left( \frac{\partial x_n}{\partial x} \right)^2 + 2 \operatorname{Sch}(x_n, x) \right] \tag{73}$$

where $x_n = f^n(x)$ is the image of $x$ after $n$-cycle driving. For numerical stability, in particular near the unstable fixed point, we invoke chain rule to reduce both terms in the above formula to a single cycle map $x_1 = f(x)$. It is worth to mention that the chain rule for Schwarzian derivative has the following interesting form

$$\operatorname{Sch}(x_n, x) = \left( \frac{\partial x_{n-1}}{\partial x} \right)^2 \operatorname{Sch}(x_n, x_{n-1}) + \operatorname{Sch}(x_{n-1}, x). \tag{74}$$

---

[15]The situation becomes different if we consider open boundary conditions. Although there are still one chiral peak and one anti-chiral peak, the open boundaries couple the chiral and anti-chiral modes together. Then there will be linearly growing entanglement between the chiral and anti-chiral peaks of energy density.

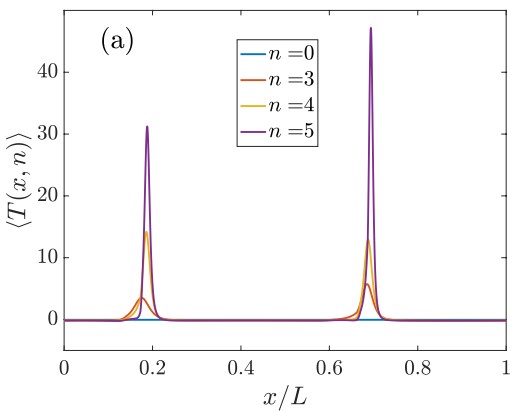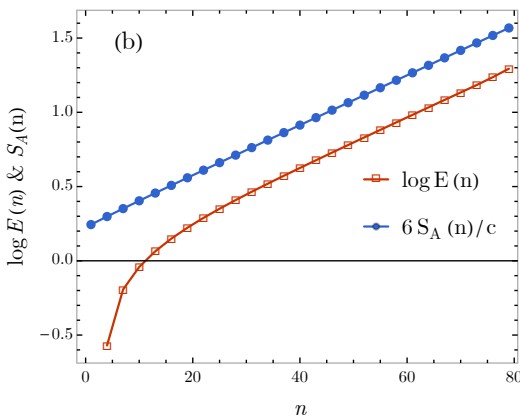

Figure 9: (a) Energy-momentum density $\langle T(x,n) \rangle$ after different numbers of driving cycles $n$. The parameters are $L = 2\pi$, $x_m = 0.45$, and $T_0/L = 0.42$, $T_1/L = 0.05$. Also see Fig. 1(b) for $T_0/L = 0.28$, $T_1/L = 0.05$, which shows three peaks. (b) Total chiral energy v.s. the entanglement entropy growth. The parameters are $L = 2\pi$, $x_m = 0.45$ and $T_0/L = 0.28$, $T_1/L = 0.05$. Results are plotted for every three cycles. (See the red curve in Fig. 2 for the entanglement entropy of full cycles, see Fig. 1 (b) for the corresponding energy-momentum density distribution)

The evolution of the entanglement entropy follows (38) and can be computed similarly. We emphasize that the energy density will become negative without including the Schwarzian term as can be seen from (73). Thus, the quantum fluctuation (i.e. the Schwarzian term, which represents the Casimir energy in free theory) plays an important role in considering the energy evolution especially from the ground state.

The results for the chiral energy-momentum density are shown in Fig. 1 (b) and Fig. 9 (a). We can see that it quickly develops peaks at positions exactly matching the unstable fixed points for the one-cycle conformal map. In contrast to the $SL_2$ case, the peaks are not of equal height. For example, in Fig. 9 (a), one peaks is significantly higher than the other. By keeping track of which peak is the higher one at a fixed time, we can see that the two peaks switches position after every cycle, which is a physical demonstration on the periodicity of fixed points (period-2 in this case). In Fig. 1 (b), there are three energy peaks plotted for every three cycles, and the relative height of each peak does not change, which implies they are period-3. One can also check that total chiral energy-momentum grows exponentially in time as shown by the blue curve in Fig. 9 (b).

The results for the spatial structure of entanglement entropy are shown in Fig. 10. In (1a) and (2a), we choose the subsystem as $A = [0, x]$. As we scan $x$ from 0 to $L$, one can find a kink structure, and the locations of kinks matches the position of energy peaks shown in Fig. 9(a) and Fig. 1(b) respectively. In Fig. 10 (1a) and (2a), we choose the subsystem to be $A = [x - \delta/2, x + \delta/2]$, with $\delta$ small enough relative to the system size. As $x$ increase from 0 to $L$, we can also observe peaks, with their position matches that of the energy peaks. These observation implies that the entanglement entropy growth only comes from the excitation accumulated at the energy-momentum peaks, which is the same as the $SL_2$ cases. Every peak share entanglement with the nearest neighbor ones.

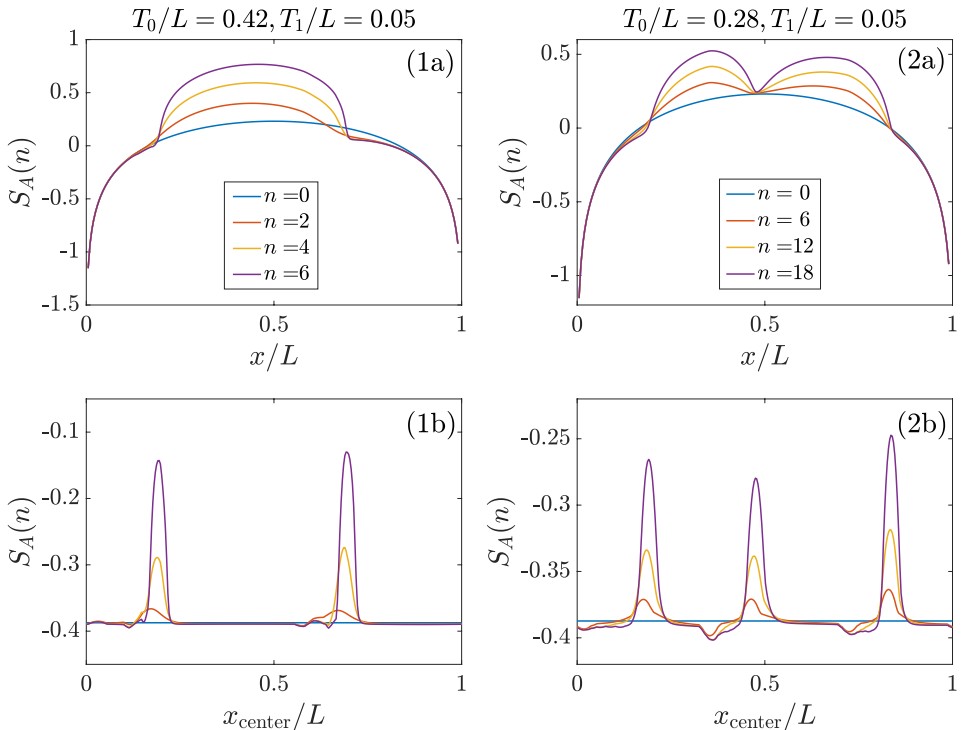

Figure 10: Entanglement entropy for fixed deformed Hamiltonian and different driving steps $n$. We choose $L = 2\pi$ and fix $x_m = 0.45$. The driving parameters are $T_0/L = 0.42$, $T_1/L = 0.05$ in (1a)(1b), and $T_0/L = 0.28$ and $T_1/L = 0.05$ in (2a)(2b), namely, the same set of parameters as used for the energy-momentum density discussion. (1a)(2a): Entanglement entropy of $A = [0, x/L]$ where $x \in [0, L]$ (1b)(2b): Entanglement entropy of $A = [x - \delta/2, x + \delta/2]$ where $x \in [0, L]$ and $\delta = L/20$. The negative value of $S_A$ in (1b) and (2b) are due to the ignorance of the UV cutoff.

We can also fix the subsystem and study the entanglement growth with respect to the cycle number $n$, the results are shown in Fig. 2 and Fig. 9(b), where the subsystem is chosen to be the left half $[0, L/2]$. When the heating phase has period-3 fixed points, the entanglement growth shows a clear oscillatory behavior within every three cycles (the red curve in Fig. 2). If we instead plot the entanglement entropy for every three cycles (the red curve in Fig. 9(b)), the result exhibits a linear growth in the late time regime. If the heating phase has period-2 fixed point, the oscillation is absent (the blue curve in  2). This is because there are only two energy peaks in this case, and we are always counting the entanglement between them no matter which peak is inside the subsystem. It is noted that the total energy and the half system entanglement shows the same growth rate as shown in Fig. 9(b), which is the same as the $\mathrm{SL}_2$ case.

### 3.4.4   Numerical simulation of free fermion on lattice

In this section, we compare the CFT results and the lattice model calculations on the entanglement entropy evolution following Ref. [38].

We consider complex free fermions on an periodic chain with only nearest neighbor hopping

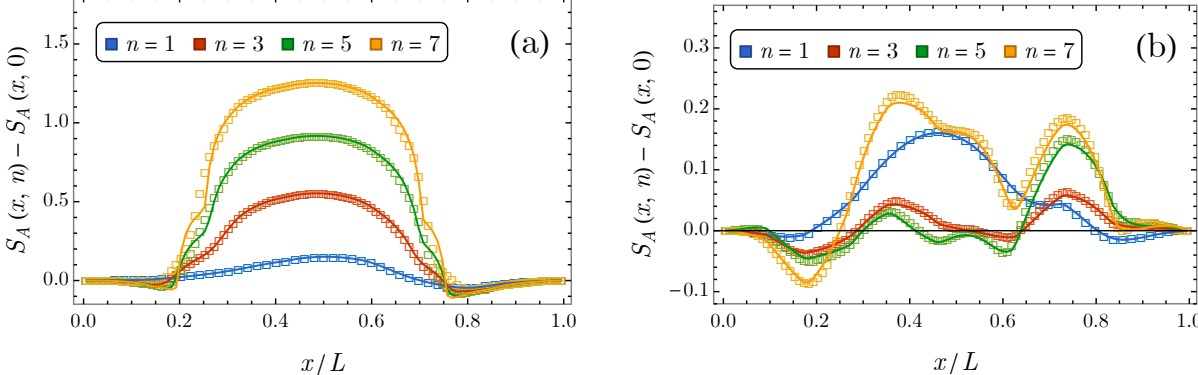

Figure 11: Comparison of the entanglement entropy evolution for $A = [0, x]$ between CFT calculations (lines) and lattice calculations (vacant squares). For all plots, we choose parameters $L = 202, x_m = 0.45$, and we subtract the initial value of the entanglement entropy to get rid of the non-universal piece. The driving parameters are chosen as (a) $T_0/L = 0.42, T_1/L = 0.05$ and (b) $T_0/L = 0.28, T_1/L = 0.05$, both of which are in the heating phase.

at the half-filling. The Hamiltonians $H_0, H_1$ for the two-step driving are

$$
\begin{cases}
H_0 = t \sum_{j=1}^{L} c_j^\dagger c_{j+1} + h.c. \\
H_1 = t \sum_{j=1}^{L} v(j)\, c_j^\dagger c_{j+1} + h.c.
\end{cases}
\quad
v(j) =
\begin{cases}
2\sin^2\left(\dfrac{\pi j}{L_A}\right), & j \leqslant L_A, \\
2\sin^2\left(\dfrac{\pi(j - L_A)}{L_B}\right), & L_A \leqslant j < L,
\end{cases}
\tag{75}
$$

where $c_j$ are fermionic operators satisfying the canonical anti-commutation relations $\{c_j, c_k^\dagger\} = \delta_{jk}$, $L = L_A + L_B$ is the number sites and $v(j)$ is the discretized deformation function. The initial state is chosen to be the ground state of $H_0$

We make stroboscopic measurement on the entanglement entropy and compare that with the CFT calculation. Fig. 1(b) shows the results for the half-system entanglement as a function of the driving periods and we find good agreement at the early time regime. However, discrepancy does appear in the late time, because high energy excitation eventually dominates the dynamics of the lattice model and is beyond the CFT description. To further support our CFT result on the spatial structure, we also examine the entanglement for the subsystem $A = [0, x]$ as a function of $x$, and the results are in Fig. 11. They show a remarkable agreement in the early time regime. Notice that both the chiral and anti-chiral components are deformed in our lattice calculation. Thus, the effect of anti-chiral deformations must be include in the CFT calculation to give a fair comparison. This accounts for the difference between Fig. 10(1a) (2a) and Fig. 11

## 4 Discussion and conclusion

In this work, we have studied the non-equilibrium dynamics in Floquet CFTs with generally deformed Hamiltonians. The time evolution of correlation functions, entanglement entropy, and energy-momentum density distribution are analyzed in detail. It is found that in the

heating phase of Floquet CFTs, the physical properties are determined by the emergent spatial fixed points of operator evolution. Compared to the $SL_2$ deformed Flouqet CFTs, there are more internal structures in the heating phases. One can have distinct heating phases with different numbers of spatial fixed points, which result in different entanglement patterns and energy-momentum density distribution.

We would like to comment on several interesting features in the Floquet CFTs and some future problems:

Despite the fact that the entanglement pattern can be directly read out from the fixed point structure, the evolution of the energy density distribution seems to be less constrained. First, the quantum effects (such as Casimir energy) are important to the growth of the energy density which makes it difficult to determine the sign of the energy-momentum density at the spatial unstable fixed point. Second, it is noted that the growth rate of the entanglement entropy is controlled by the stable fixed point while the growth rate of energy density is mainly controlled by the unstable ones. And the relation between them is not clear to us. Same questions can also be raised in the context of a global quench generated by generally deformed Hamiltonians. We leave these for future study.

In this work, we use the topological surface operator to give a general definition of the deformed Hamiltonian. Such a formulation automatically suggests a generalization to higher dimensional CFTs. It is noted the conformal group in higher dimensional CFT is $SO(d+1, 1)$, $d > 2$ (See the discussion in Sec.2), which is non-compact. Thus we expect there are still 'heating' and 'non-heating' phases in the Floquet driving. It will be interesting to study the detailed features of related quantities such as the entanglement entropy and energy density evolution in the higher dimensional cases.

Our approach can be straightforwardly generalized to more general driving sequences such as the quasi-periodic and random driving, which is shown to yield more interesting features in the $SL_2$ deformed case compared to the periodic one [43]. In the generalized Flouqet CFTs, as we extend the $\mathfrak{sl}_2$ algebra to the Virasoro algebra, it is also interesting to enquire the phase diagram for quasi-periodic and random driving.

It is also interesting to consider the Floquet drivings with non-unitary time evolution. In this case, more general conformal mappings are allowed in the operator evolution. It is expected that more rich patterns of entanglement/energy evolution can be observed. Some initial efforts along this direction can be found in Ref. [60].

In addition, since our results also hold for the large-$c$ CFT, it is desirable to study the holographic dual of our setup. It is also interesting to compare our setup to the Floquet setups as studied in $AdS_4/CFT_3$ in Ref. [61,62].

*Note added:* During the preparation of this work, we noted a related work [63], which also studies Floquet CFTs beyond the $SL_2$ deformation, while concrete examples are different. We thank the authors for their communications.

# 5 Acknowledgement

RF, XW, and AV are supported by a Simons Investigator award (AV) and by the Simons Collaboration on Ultra-Quantum Matter, which is a grant from the Simons Foundation (651440, AV). AV and RF are supported by the DARPA DRINQS program (award D18AC00033). This

work was also partially supported by Gordon and Betty Moore Foundation's EPiQS initiative through Grant No. GBMF4303 at MIT (XW). YG is supported by the the Simons Foundation through the "It from Qubit" program.

# A    Quantum quenches with general deformation

Our discussion in the main text can be straightforwardly applied to the quantum quench problem, with the quenched Hamiltonian in the form of (28)

$$H_v = \int_0^L \frac{dx}{2\pi} \left( v(x)T(x) + \overline{v}(x)\overline{T}(x) \right) \tag{76}$$

In the quantum quench, for simplicity, we choose the initial state as the ground state of a homogeneous CFT with $v(x) = \overline{v}(x) = 1$ in the above Hamiltonian. Then at $t = 0$, we quench the Hamiltonian to the non-homogeneous one with smooth $(v(x), \overline{v}(x))$, and evolve the system in time. [16] For simplicity of discussion, we only deform the chiral part of Hamiltonian and keep $\overline{v}(x) = 1$.

The quench dynamics depends on whether there are zeros in the deformation function $v(x)$. If there are no zeros, then every physical observable will be a periodic function of time, with the period $T = L_{\text{eff}} = \int_0^L \frac{dx}{v(x)}$, which is the effective length of the total system after deformation. That means the entanglement entropy will oscillate in time. If there are zeros in $v(x)$, the entanglement entropy will in general grow in time. As shown in (77) is an example of the profile $v(x)$. In our convention, the chiral operators move right for $v(x) > 0$ and move left for $v(x) < 0$. Then one can find that the zeros of $v(x)$ correspond to the stable ($\bullet$) and unstable ($\circ$) fixed points in the operator evolution

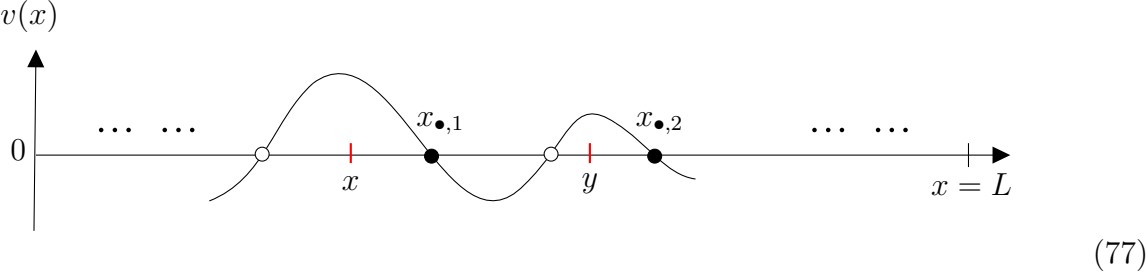

$$\tag{77}$$

For smooth $v(x)$, similar to the discussion in Sec. 3.3.1, the entanglement entropy of $A = [x, y]$ depends on how we choose the subsystem. As shown in (77), one non-trivial choice of subsystem $A$ is that there is at least one of the unstable fixed points inside $A$. Then, following the discussion in Sec.3.3.1, we find that if $v'(x_\bullet) \neq 0$, then the entanglement entropy grows in time $t$ as

$$S_A(t) \simeq \frac{c}{12} \cdot (\lambda_{\bullet,1} + \lambda_{\bullet,2}) \cdot t, \tag{78}$$

where $\lambda_{\bullet,j} := |v'(x_{\bullet,j})|$. While if the first derivative vanishes, and we assume the leading power in the Taylor expansion is proportional to $(x - x_{\bullet,j})^{n_{\bullet,j}}$ where $n_{\bullet,j} > 1$, then we have

$$S_A(t) \simeq \frac{c}{12} \left( \frac{n_{\bullet,1}}{n_{\bullet,1} - 1} + \frac{n_{\bullet,2}}{n_{\bullet,2} - 1} \right) \log t. \tag{79}$$

---

[16]For related discussions on quantum quenches with inhomogeneous Hamiltonians, one can also refer to, e.g., Refs. [64–66].

Here we have neglected the finite constant terms in Eqs.(78) and (79).

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
