# Peer review of "Floquet conformal field theories with generally deformed Hamiltonians"

_SciPost Physics_

## Round 2 · Referee Report · Ramasubramanian Chitra (Referee 1) · 2021-1-26

Strengths

  1. The generalisation to smooth deformations in Floquet CFT theories is an important step in the Floquet CFT approached pioneered by one of the authors and deserves being published.

  2. Detailed methodology

  3. Comparison with lattice model calculations of relevance to condensed matter systems.

Weaknesses

  1. Uneven writing which makes it hard to read at times.

  2. Many typos/small grammatical errors.

Report

The authors generalise the results obtained by some of them earlier
for SL(2,C) deformed hamiltonians to the case of more general
deformations. To do this, they use a geometric approach which helps
map the problem of the driving to dynamical maps and general
conformal transformations. They also present analytic results for
the the energy as well as the entanglement entropy.
As a test case, they derive explicit results for a model with a
specific kind of deformation and also present comparisons to
lattice systems. This generalisation to smooth deformations is an important step in
the Floquet CFT approached pioneered by one of the authors and
deserves being published.

The authors strive to present their methodology in a detailed manner.
However, the quality of the writing in this paper is rather
uneven (some sections are clearly written while some are not) which makes
it hard reading at times. The paper unfortunately abounds with many grammatical errors
and the authors should take the time to improve the text of their
manuscript. To conclude, provided the authors improve their text and clearly
address the points raised below and incorporate their answers in the manuscript, I would recommend publication in SciPost.

Requested changes

I do have questions and comments on the manuscript:

In particular, I would recommend that the authors reorganise the introduction as it toggles between generalities and highly detailed points. The introduction as is seems to be addressed to the small community of researchers in this narrow field rather than a wider community as assumes a fairly detailed knowledge of their results in earlier publications. Trying to place the problem within a wider context will help the readability of the manuscript.

  1. In the very second paragraph:

" Second, it can be rephrased as a topological surface operator associated with conformal Killing vectors, which makes the generalisation to higher dimension transparent............... " This entire paragraph is a short and technical paraphrase of the work done in the next 30 pages of the manuscript and should be simplified/improved.

  1. On page 3, the authors use "people" to refer to earlier works. This informal expression should be avoided.

  2. Both points 2 and 3 in "Concrete examples" should be explained in a clear and accessible manner keeping in mind that this is still an introductory section. As written, the reader cannot comprehend at this stage what p-cycles are (let alone the specific 3 and 6 cycle cases referred).

  3. Same goes for the point 3:

"These energy-density peaks also share entanglement with its nearest neighbors. A cartoon is shown in Fig. 1(c). These energy peaks can be understood as an accumulation of particles that come from some source, denoted by the red dots on top of the vacant dots. Each dot is entangled with its partner while they departure towards different destinations, which gives rise to the energy peaks as well as their entanglement."

This text is quite incomphrehensible and needs to be completely reworked.

  1. Page 4 Grammatical error (if exist)

  2. On page 5, in point 3 is it necessary to have "highly excited states" as opposed to merely "excited states" ?
    For example, such behaviour seen even if initial state is the ground state of the open undeformed system as in the papers of the last author of the present manuscript.

  3. Page 6, typo: A conformal field theory at the space-time dimension - "in" spacetime...

  4. Page 11: typo after sec 2.5 recipe ....receipt

  5. Considering these deformed hamiltonians, how are their spectra modified ? I refer to the literature on SSD hamiltonians where dipolar quantisations were used to demonstrate that in some limits, the Moebius limit, the spectra of the SSD can become continuous. To my knowledge this latter aspect was a crucial ingredient for the transition to the heating phase. Can the authors discuss the spectra of the system with smooth general deformations in this context and their role in understanding the heating phase diagram ?

  6. Eq.35 Misleading notation H_v_n (v_n is also Fourier components of v(x) in space cf.Eq.28) This needs to be fixed in a consistent manner in the later sections too.

  7. Page 17.. point 3 typo merge ..merges

  8. In the discussion of the entanglement entropy, the authors refer to figures 1 multiple times. The authors should avoid this kind of going back and forth across 15 pages to explain their results. As this is a long paper, the authors should consider whether figure 1 can be modified so that some of the plots can be moved to this section. This will strongly facilitate the readability of this section.

  9. The authors introduce the variable lambda for the different fixed points. Are these just the Lyapunov exponents ? If yes, this terminology could be used in the text.

  10. In Eq.66, where one of the points x,y is at an unstable fixed point, can lambda_0 1 = \lambda_dot 2 ? If yes, this would mean a constant entanglement entropy in the heating phase which is confusing. Can the authors elaborate on this? A discussion of the multiple scenarios presented will be useful.

  11. Figure 8: Apart from the limiting cases of x_m=0,0.5, the full Virarsoro algebra comes into play so naively one could expect the phase diagrams to be similar, but this clearly is not the case. Can the authors present a physical explanation as to why x_m affects the phase diagram dramatically?

The phase diagrams plotted in Fig.8 are very reminiscent of those obtained with quasiperiodic driving with fractal like structures in some of the cited references. Is this merely accidental ? In this plot, it is not clear if the paler blue regions are heating or non heating regions. The figure caption could be improved. What are the qualitative/quantitative differences between these lighter blue slivers ? Is there a simple physical picture which explains this ? The authors should devote some space to describing the fractal like structures and what the physical differences are.

  1. In general, how does periodicity of fixed points affect EE ? Can one characterise the transition between heating regimes with different numbers of fixed points ? Does the prefactor of the EE jump across this or are there other signatures ? In the sense of a dynamical map, what kind of bifurcations do these boundaries correspond to ?

  2. In the caption of Fig.9, the authors again reference a comparison with Fig.1 which corresponds to a different point in parameter space. It would make sense to have these figures side by side in Fig.9 rather than 20 odd pages apart.

Also the figures in this plot correspond to x_m=0.45. It would make sense to have the phase diagram corresponding to this value of x_m in the the collection of phase diagrams in Fig. 8. Additionally, I would recommend that the authors add a couple of figures which show the difference in these observables for heating regimes with q=1 and q\neq 1

  1. The authors should also add a discussion about micromotion for the physical examples studied. The authors briefly mention this, but a more in-depth discussion for the model example is really warranted for a long paper as this.

  2. Can one obtain the effective Floquet hamiltonian using this geometrical approach ? This point should also be addressed in the current paper.

  3. There are a multitude of small grammatical errors in the paper which I have not listed.

  • validity: top
  • significance: good
  • originality: high
  • clarity: good
  • formatting: -
  • grammar: acceptable

Author:  Ruihua Fan  on 2021-02-09  [id 1213]

(in reply to Report 1 by Ramasubramanian Chitra on 2021-01-26)

We thank the referee for the careful reading and overall positive feedback, and, in particular, very helpful questions and suggestions. We tried to improve our manuscript based on referee's suggestions. Here is a list of changes : 1) Parts of the introduction are rewritten to improve the readability, such as the three beginning paragraphs (between the two red lines), and paragraphs in red. 2) In Fig 8, the original phase diagram with x_m=0.4 is replaced with a new diagram with the parameter x_m=0.45, which is also the parameter used by later figures. 3) Several discussions are added to Sec. 2, 3, to improve the readability and answer some of referees questions. 4) Typos and grammatical errors are fixed.

More detailed answer to each questions can be found below. And the new version has also been uploaded.

* Reply to each questions * (Referee's questions are indicated by ">", our answers by "A:")

>I do have questions and comments on the manuscript:

>In particular, I would recommend that the authors reorganise the introduction as it toggles between generalities and highly detailed points. The introduction as is seems to be addressed to the small community of researchers in this narrow field rather than a wider community as assumes a fairly detailed knowledge of their results in earlier publications. Trying to place the problem within a wider context will help the readability of the manuscript.

A: We have rewritten the introduction part and we hope that it will be satisfying.

>1. In the very second paragraph:

" Second, it can be rephrased as a topological surface operator associated with conformal Killing vectors, which makes the generalisation to higher dimension transparent............... " This entire paragraph is a short and technical paraphrase of the work done in the next 30 pages of the manuscript and should be simplified/improved.

A: We have rewritten this part to move the technical part to later discussion.

>2. On page 3, the authors use "people" to refer to earlier works. This informal expression should be avoided.

A: We have rewritten this part.

>3. Both points 2 and 3 in "Concrete examples" should be explained in a clear and accessible manner keeping in mind that this is still an introductory section. As written, the reader cannot comprehend at this stage what p-cycles are (let alone the specific 3 and 6 cycle cases referred).

A: We have added a few sentences to explain what “cycle” refers to.

>4. Same goes for the point 3:

>"These energy-density peaks also share entanglement with its nearest neighbors. A cartoon is shown in Fig. 1(c). These energy peaks can be understood as an accumulation of particles that come from some source, denoted by the red dots on top of the vacant dots. Each dot is entangled with its partner while they departure towards different destinations, which gives rise to the energy peaks as well as their entanglement."

>This text is quite incomphrehensible and needs to be completely reworked.

A: We have rewritten this paragraph.

>6. On page 5, in point 3 is it necessary to have "highly excited states" as opposed to merely "excited states" ? For example, such behaviour seen even if initial state is the ground state of the open undeformed system as in the papers of the last author of the present manuscript.

A: We need this assumption (highly excited) for the analytical argument to work. For generic initial states, we do not have a good way to bound the local energy density, as we discussed in Sec.3.3.2. The previous study focuses on the SL2 case, which does not have this subtlety.

>13. The authors introduce the variable lambda for the different fixed points. Are these just the Lyapunov exponents ? If yes, this terminology could be used in the text.

A: For the unstable fixed points (Eq 59), lambda characterizes the instability and is the Lyapunov exponent for the map f. While for the stable fixed point (Eq 58), lambda describes how fast a point flows to the stable fixed point and is not the Lyapunov exponent for the map f. We want to treat the two kinds of fixed points on an equal footing, that is why we do not introduce this terminology.

>14. In Eq.66, where one of the points x,y is at an unstable fixed point, can lambda_0 1 = lambda_dot 2 ? If yes, this would mean a constant entanglement entropy in the heating phase which is confusing. Can the authors elaborate on this? A discussion of the multiple scenarios presented will be useful.

A: Yes, this is correct. Colloquially speaking, the source happens to balance the sink, which results in a constant entanglement. We add one sentence to comment on this scenario.

>15. Figure 8: Apart from the limiting cases of x_m=0,0.5, the full Virarsoro algebra comes into play so naively one could expect the phase diagrams to be similar, but this clearly is not the case. Can the authors present a physical explanation as to why x_m affects the phase diagram dramatically?

A: By changing x_m, we simply change the weight of Fourier components in the driving Hamiltonian, therefore the phase diagram does change continuously as a function of x_m (which may be not apparent by simply looking at the phase diagram at those discrete x_m we plotted) However, the area of each region is non-generic and it does not necessarily change with x_m linearly. In addition, these figures share the same qualitative features, e.g. heating phases with higher periodicity, all heating phases terminate at some rational point of the real axis, and so on. For some choice of x_m, it is numerically hard to compute heating phases with large numbers of fixed points to high precision because their heating rate is generally small and might make the diagram look different, which is merely an artifact. Thus, we respectfully disagree that x_m affects the phase diagram dramatically.

>The phase diagrams plotted in Fig.8 are very reminiscent of those obtained with quasiperiodic driving with fractal like structures in some of the cited references. Is this merely accidental ?

A: Indeed, the phase diagram here is quite similar to the phase diagram with quasi-periodic driving approximated by a periodic one. In the quasi-periodic limit, the measure of the non-heating phase goes to zero, while in the generalized Floquet driving case, the measure of the non-heating phase is still nonzero. This is an important difference.

>In this plot, it is not clear if the paler blue regions are heating or non heating regions. The figure caption could be improved. What are the qualitative/quantitative differences between these lighter blue slivers ?

A: Lighter blue slivers have a larger number of fixed points. We add one sentence in the caption of Fig.8 to be more clear.

>Is there a simple physical picture which explains this? The authors should devote some space to describing the fractal like structures and what the physical differences are.

A: As for the structure of the phase diagram of the generalized Floquet driving itself, we do not have a proof of whether the phase diagrams are fractal or not. But the structure of the phase diagrams here is related to the Arnold tongue in the study of circle maps.

>16. In general, how does periodicity of fixed points affect EE ?

A: The periodicity of fixed points will affect the fine structure of EE growth. For example, suppose the periodicity of fixed points is “q”, then the EE will grow monotonically after every “q” driving periods. But there can be fine structures such as oscillations within the “q” periods, as seen in fig.2. In the numerical calculation of our concrete model, we also found that heating phases with a higher periodicity of fixed points generally have smaller heating rates. But we do not have any analytical proof of this statement.

>Can one characterise the transition between heating regimes with different numbers of fixed points ? Does the prefactor of the EE jump across this or are there other signatures ?

A: In general, there are indeed interesting features of EE evolution at the transitions. On the one hand, certain stable and unstable fixed points will merge into critical fixed points at the transition. As discussed around Eq.69, if the subsystem is chosen to contain only these fixed points, then the EE will exhibit logarithmic growth. On the other hand, the rest fixed points may stay stable/unstable. If the subsystem is chosen to contain these fixed points, we may still see linear growth in EE.

>In the sense of a dynamical map, what kind of bifurcations do these boundaries correspond to ?

There are two transitions. Both are certain saddle-node bifurcation, in the sense that pairs of higher-period fixed points are annihilated across the boundary. This is also observed in the study of circle maps and Arnold tongues. [e.g. see Commun. Math. Phys. 106, 353-381 (1986) “Bifurcations of Circle Maps:ArnoPd Tongues, Bistability and Rotation Intervals” ]

One is from one heating phase to the non-heating phase . This corresponds to a saddle-node bifurcation of the map f^p, where p denotes the period of the heating phase. The other is the transition between heating phases with different periods and numbers of fixed points. Let us assume that the number of fixed points equals the period. Let q1, q2 denote the number of fixed points of the two nearby phases, then the transition can be viewed as a saddle-node bifurcation of the map f^{q1times q2}.

>12. In the discussion of the entanglement entropy, the authors refer to figures 1 multiple times. The authors should avoid this kind of going back and forth across 15 pages to explain their results. As this is a long paper, the authors should consider whether figure 1 can be modified so that some of the plots can be moved to this section. This will strongly facilitate the readability of this section.

>17. In the caption of Fig.9, the authors again reference a comparison with Fig.1 which corresponds to a different point in parameter space. It would make sense to have these figures side by side in Fig.9 rather than 20 odd pages apart.

A: While we appreciate the referee's valuable feedback, we intended to do so in order to highlight some of our results for readers who do not want to read the detailed discussion. We have rewritten the introduction part and hope that the referee would find it satisfying.

>Also the figures in this plot correspond to x_m=0.45. It would make sense to have the phase diagram corresponding to this value of x_m in the the collection of phase diagrams in Fig. 8.

A: We have replaced the diagram of x_m=0.4 with one of x_m=0.45.

>Additionally, I would recommend that the authors add a couple of figures which show the difference in these observables for heating regimes with q=1 and qneq 1

A: We have added one paragraph to discuss this at the end of Sec. 3.4.3

>18. The authors should also add a discussion about micromotion for the physical examples studied. The authors briefly mention this, but a more in-depth discussion for the model example is really warranted for a long paper as this.

A: We have added a concrete discussion in the introduction when explaining Fig 1 (b) and Fig 2.

>9. Considering these deformed hamiltonians, how are their spectra modified ? I refer to the literature on SSD hamiltonians where dipolar quantisations were used to demonstrate that in some limits, the Moebius limit, the spectra of the SSD can become continuous. To my knowledge this latter aspect was a crucial ingredient for the transition to the heating phase. Can the authors discuss the spectra of the system with smooth general deformations in this context and their role in understanding the heating phase diagram ?

>19. Can one obtain the effective Floquet hamiltonian using this geometrical approach ? This point should also be addressed in the current paper.

A: The spectral of the effective Hamiltonian, which is related to the one of the deformed Hamiltonian, does play an important role in determining the phase diagram. However, this is more along the line of solving the problem from an algebraic point of view, and an analytical calculation given a generic driving Hamiltonian is hard and it is an open question to us. In this manuscript, we prefer to highlight the geometric viewpoint, therefore we do not attempt to do this.

>10. Eq.35 Misleading notation H_v_n (v_n is also Fourier components of v(x) in space cf.Eq.28) This needs to be fixed in a consistent manner in the later sections too.

A: Thanks for pointing it out. We have added tilde to v_n in Eq 26 and 28 to avoid confusion.

>5. Page 4 Grammatical error (if exist) >7. Page 6, typo: A conformal field theory at the space-time dimension - "in" spacetime... >8. Page 11: typo after sec 2.5 recipe ....receipt >11. Page 17.. point 3 typo merge ..merges >20. There are a multitude of small grammatical errors in the paper which I have not listed.

A: Thanks for pointing these out. We have corrected these typos as well as some others we have found.

---

## Round 2 · Referee Report · Anonymous (Referee 2) · 2021-2-8

Strengths

This paper deals with the Floquet dynamics in CFT in 1+1 dimensions.
This is a recent topic in the study of out-of-equilibrium phenomena that is attracting a lot of interest. One of the reasons is that one can use the powerful techniques of CFT to study in great detail the dynamics and in some cases even derive exact analytical results.

The authors generalize in this paper the results obtained previously in reference [43] concerning sl(2) deformations of the driving Hamiltonian to general inhomogeneous models. They present the general formalism in terms of a simple geometric interpretation and focus their attention on the distribution of the energy-momentum tensor and the entanglement entropy for different cuts. The behaviour of these quantities in explained in terms of the stroboscopic trajectories that has non fixed points, stable and unstable fixed points and critical fixed points that lead to the phases denoted, non heating, heating and critical. This picture was obtained previously but here the phase diagram is much reacher and has a sort of fractal structure. The results have also a nice interpretation in terms of the Cardy-Calabrese quasiparticle picture that seems to hold also for this class on models. Finally the authors present a numerical analysis using a Dirac fermion driven with a lattice version of the continuum model analysed previously. The agreement between the CFT prediction and the numerical one is very good except in some regimes where lattice effects become important. The paper is very well written and contains very interesting new results that can be applied to other inhomogenuous models.

Weaknesses

A topic that is not discussed is the spectrum of the Floquet Hamiltonian, that is the quasi energies. Would it throw further information on these models? The entanglement spectrum is not also discussed but the entanglement entropy already explains the physics involved.

Report

In my opinion the paper can be published in its present form.
  • validity: high
  • significance: high
  • originality: high
  • clarity: good
  • formatting: excellent
  • grammar: good

Author:  Ruihua Fan  on 2021-02-09  [id 1214]

(in reply to Report 2 on 2021-02-08)

We acknowledge referee's accurate summary of our work and approval for publication. We also thank referee's interesting questions.

The quasi-spectrum of the Floquet Hamiltonian and the entanglement spectrum should be doable in the SL2 driving case, and the former was already discussed before. In this general case, both questions are very interesting but also technically hard, and are open questions to us.

---

## Editorial Decision

resubmitted